# Host immunity modulates the efficacy of microbiota transplantation for treatment of *Clostridioides difficile* infection

Eric R. Littmann[1], Jung-Jin Lee[2], Joshua E. Denny[3], Zahidul Alam[3], Jeffrey R. Maslanka[3], Isma Zarin[3], Rina Matsuda[4], Rebecca A. Carter[5], Bože Susac [5], Miriam S. Saffern [5], Bryton Fett[2], Lisa M. Mattei [2], Kyle Bittinger [2] & Michael C. Abt [3✉]

Fecal microbiota transplantation (FMT) is a successful therapeutic strategy for treating recurrent *Clostridioides difficile* infection. Despite remarkable efficacy, implementation of FMT therapy is limited and the mechanism of action remains poorly understood. Here, we demonstrate a critical role for the immune system in supporting FMT using a murine *C. difficile* infection system. Following FMT, *Rag1* heterozygote mice resolve *C. difficile* while littermate *Rag1*[−/−] mice fail to clear the infection. Targeted ablation of adaptive immune cell subsets reveal a necessary role for CD4[+] Foxp3[+] T-regulatory cells, but not B cells or CD8[+] T cells, in FMT-mediated resolution of *C. difficile* infection. FMT non-responsive mice exhibit exacerbated inflammation, impaired engraftment of the FMT bacterial community and failed restoration of commensal bacteria-derived secondary bile acid metabolites in the large intestine. These data demonstrate that the host's inflammatory immune status can limit the efficacy of microbiota-based therapeutics to treat *C. difficile* infection.

[1] The Duchossois Family Institute, University of Chicago, Chicago, IL, USA. [2] Division of Gastroenterology, Hepatology, and Nutrition, Children's Hospital of Philadelphia, Philadelphia, PA, USA. [3] Department of Microbiology, Perelman School of Medicine, University of Pennsylvania, Philadelphia, PA, USA. [4] Department of Pathobiology, University of Pennsylvania School of Veterinary Medicine, Philadelphia, PA, USA. [5] Lucille Castori Center, Molecular Microbiology Core Facility, Sloan-Kettering Institute, Memorial Sloan Kettering Cancer Center, New York, NY, USA. ✉email: Michael.abt@pennmedicine. upenn.edu

The microbiota, consisting of the community of bacterial, viral, fungal, and protozoa organisms that inhabit a mammalian host, can impact susceptibility to a range of diseases including cancer, diabetes, allergy, obesity, and infection[1–4]. An altered or dysbiotic intestinal microbiota is observed in all of these disease states suggesting that resetting the intestinal microbiota with a composition of bacteria from healthy individuals could treat disease[5,6]. This theory is supported by animal studies demonstrating alleviation of disease through transplantation of bacteria into the diseased host's intestinal tract[7–12]. Significant advancements have been made in defining the community of microbial species that are important in supporting human health[11–18]. Translating these findings into viable microbiota-based therapies has had limited success in the clinic, however, and is not yet considered a treatment option for most diseases impacted by microbial dysbiosis[19]. One notable exception is the use of fecal microbiota transplantation (FMT) to treat recurrent *Clostridioides difficile* infection[20]. FMT treatment of *C. difficile* infection is the first microbiota-based therapy clinically proven to ameliorate disease[21]. Before such treatment strategies can be broadly implemented, the underlying principles determining success for microbiota-based therapy need to be elucidated.

*C. difficile*, a gram positive, spore-forming, obligate anaerobe is currently the most common nosocomial infection encountered by hospitalized patients, with nearly a half million patients infected and an estimated 13,000–30,000 deaths annually in the USA alone[22–24]. This opportunistic pathogen infects the large intestine following perturbation of the intestinal microbiota. Vancomycin or fidaxomicin are first line treatment options that effectively target the vegetative form of *C. difficile*, but fail to resolve the underlying condition that promotes infection, a dysbiotic microbiota[25]. Recurrence of *C. difficile* infection due to repeated failed antibiotic treatments is estimated as high as 25–35% and can cause potentially lethal fulminant colitis[22,26,27]. The high recurrence rate following standard antibiotic treatment has spurred the development of new treatment modalities such as FMT. In 2013, the first controlled, double-blinded study reported the superior efficacy of FMT compared to antibiotics in treatment of recurrent *C. difficile* infection[21]. Subsequent studies demonstrate FMT has an 80–90% cure rate in individuals and this treatment option is now incorporated as standard care for recurrent disease in both Europe and the USA[28–30].

Intestinal dysbiosis in *C. difficile*-infected patients is characterized by a loss of bacterial diversity and altered production of microbiota-derived intestinal metabolites[31,32]. Successful FMT restores homeostasis both in the intestinal microbial community and metabolome[33,34]. For example, following FMT, commensal bacteria that convert host-derived primary (1°) bile acids into secondary (2°) bile acids repopulate the large intestine and 2° bile acid pools are restored to levels observed in healthy individuals[35]. The presence of 2° bile acid converting bacteria promotes colonization resistance against *C. difficile*[13] and restoration of the 2° bile acid pools is associated with successful FMT in recurrent *C. difficile*-infected patients[34–36]. These reports provide evidence that efficacy of the FMT is dependent on the bacterial consortium successfully engrafting in the intestine of the infected host to restore the intestinal microenvironment to pre-infection conditions and create an inhospitable environment for *C. difficile*. The role of host immune factors in supporting FMT engraftment and subsequent clearance of *C. difficile* has not been explored and is the focus of this report.

In this study, we demonstrate that multiple strains of immunodeficient mice, all of which lack CD4$^+$ Foxp3$^+$ T-regulatory (T$_{reg}$) cells, exhibit increased intestinal inflammation compared to immunocompetent mice when persistently infected with *C.*

*difficile* and fail to resolve infection following FMT. The transplanted bacteria from the FMT inoculum do not completely engraft in the large intestine of FMT non-responsive mice and the intestinal metabolite profile is not restored to pre-infection levels. These data reveal an important role for the host immune system in supporting bacterial engraftment and subsequent resolution of *C. difficile* infection.

## Results

**FMT mediated clearance of *C. difficile* infection resolves intestinal inflammation.** Fecal microbiota transplantation is a proven treatment for recurrent *C. difficile* disease[37], however, whether the host's immune system contributes to FMT-mediated resolution of *C. difficile* is unknown. To address this question, we first established a murine model of *C. difficile* infection followed by FMT treatment (Fig. 1a). Similar to previous reports in both mice and humans[7,21,38], FMT reduced *C. difficile* burden and toxin levels in the large intestine of persistently infected C57BL/6 mice to below the limit of detection while sham PBS treatment did not impact *C. difficile* burden (Supplementary Fig. 1A,B). By day 10 post-FMT, recipient mice had resolved *C. difficile* infection-driven intestinal inflammation characterized by immune cell infiltration, submucosa edema (Supplementary Fig. 1C), large intestine crypt elongation (Supplementary Fig. 1D), and elevated mRNA expression of proinflammatory genes in the proximal colon (Supplementary Fig. 1E). These data establish that FMT can indirectly shape intestinal immune homeostasis via resolution *C. difficile* infection and provoke further investigation into the role of the immune system in supporting FMT-mediated *C. difficile* resolution.

**Immunodeficient *Rag1*$^{-/-}$ mice exhibit impaired resolution of *C. difficile* infection following FMT.** If FMT resolves *C. difficile* infection independently of the host immune system, FMT therapy should be equally successful in immunodeficient hosts. To test this null hypothesis and to begin to dissect the role of the immune system in supporting a FMT, T and B cell deficient *Rag1*$^{-/-}$ mice and littermate control, *Rag1* heterozygous mice (*Rag1*$^{HET}$) were compared using our *C. difficile* infection followed by FMT experimental system (Fig. 1a). In agreement with previous reports[39–41], *Rag1*$^{HET}$ and cohoused *Rag1*$^{-/-}$ mice exhibited similar weight recovery from acute *C. difficile* disease (Fig. 1b) and both groups established a comparable persistent *C. difficile* infection in the large intestine (Supplementary Fig. 2A). Following establishment of persistent infection, *Rag1*$^{HET}$ and *Rag1*$^{-/-}$ mice were administered a FMT, separated into individually housed cages after FMT to prevent microbial cross contamination, and *C. difficile* was monitored in the feces. *Rag1*$^{HET}$ mice resolved *C. difficile* infection following FMT and had no detectable *C. difficile* toxin present in its cecal content (Fig. 1c, d). In contrast, *C. difficile* burden and toxin titers persisted in the large intestine of FMT-treated *Rag1*$^{-/-}$ mice thus refuting our null hypothesis (Fig. 1c, d). Histological examination of the cecum of *C. difficile*-infected *Rag1*$^{HET}$ and *Rag1*$^{-/-}$ mice demonstrated pronounced submucosa edema, cellular infiltration, and crypt elongation in *Rag1*$^{-/-}$ mice compared to *Rag1*$^{HET}$ mice (Fig. 1e). Following FMT, intestinal inflammation was resolved in *Rag1*$^{HET}$ mice but not in FMT-treated *Rag1*$^{-/-}$ mice (Fig. 1e, f). These observations suggest that the immune status of the FMT recipient is an important determinant in the ultimate success of FMT therapy.

Two potential confounding factors to this observation are distinct microbial exposure following FMT due to individually housing recipient mice following FMT and the virulence of the *C. difficile* strain used. Therefore, we assessed the efficacy of FMT in wild-type C57BL/6 and *Rag1*$^{-/-}$ mice that remained cohoused

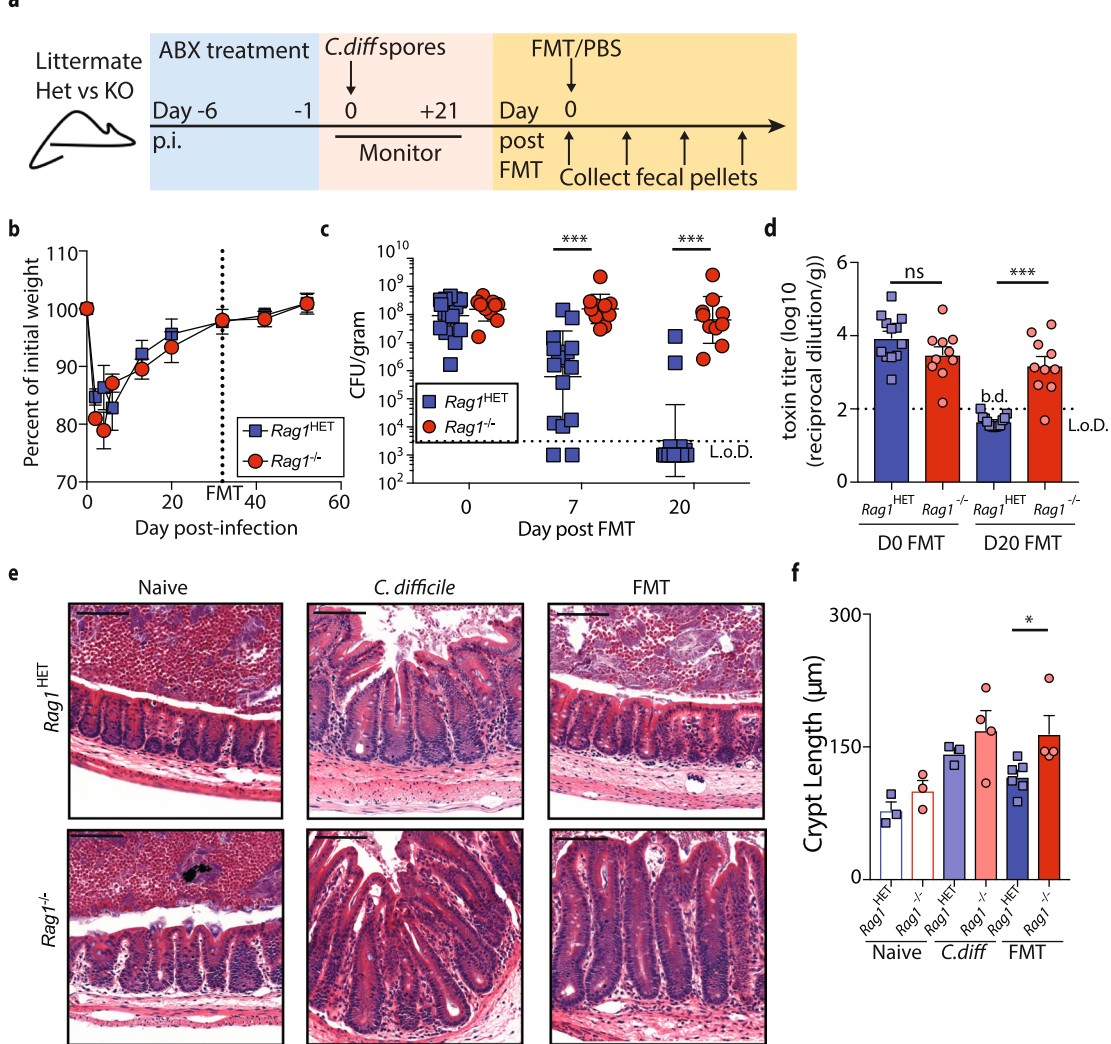

**Fig. 1 Immunodeficient *Rag1*⁻/⁻ mice exhibit impaired resolution of *C. difficile* infection following FMT. a** Schematic of antibiotic (ABX) treatment, *C. difficile* infection, and FMT. **b** Weight loss following infection in littermate, antibiotic-treated *Rag*HET (n = 6) and *Rag1*⁻/⁻ (n = 5) mice. Data is representative of three independent experiments. Data are presented as mean values ± SEM. **c** *C. difficile* burden and **d** toxin B levels in the fecal pellets of *Rag1*⁻/⁻ (n = 10) and *Rag1*HET (n = 14) mice following FMT. Data is a combination of three independent experiments. Statistical significance was calculated by a two-sided unpaired *t*-test. ***p < 0.0001. Data are presented as mean values ± SEM. **e** H&E stained cecal tissue sections and **f** mean crypt length from naïve, *C. difficile*-infected PBS treated, or *C. difficile*-infected FMT-treated *Rag1*HET and *Rag1*⁻/⁻ mice. n = 3 naïve *Rag1*HET and *Rag1*⁻/⁻ mice; n = 3 *C. difficile*-infected PBS treated *Rag1*HET mice; n = 4 *C. difficile*-infected PBS treated *Rag1*⁻/⁻ mice; n = 6 *C. difficile*-infected FMT-treated *Rag1*HET mice; n = 4 *C. difficile*-infected FMT-treated *Rag1*⁻/⁻ mice. Scale bar = 100 μm. Data is representative of two independent experiments. Statistical significance was calculated by a two-sided unpaired Mann–Whitney test; *p = 0.01 Data are presented as mean values ± SEM. b.d. below detection.

throughout the course of the experiment using both the CD196 and highly pathogenic VPI10463 *C. difficile* strains[42]. Despite cohabitation and continuous microbial exposure from FMT-treated C57BL/6 mice, FMT-treated *Rag1*⁻/⁻ mice exhibited significantly higher *C. difficile* burden (Supplementary Fig. 2B, D) and toxin titers in the feces (Supplementary Fig. 2C, E) compared to C57BL/6 mice in the context of either a CD196 strain infection (Supplementary Fig. 2B, C) or VPI10463 strain infection (Supplementary Fig. 2D, E). These data demonstrate that the differential outcome following FMT in wild-type and *Rag1*⁻/⁻ mice is independent of *C. difficile* strain or cohousing conditions.

***C. difficile*-infected *Rag1*⁻/⁻ mice exhibit exacerbated intestinal inflammation compared to *Rag1*HET mice despite similar microbiota composition.** We next analyzed the intestinal microbiota composition of *Rag1*HET and *Rag1*⁻/⁻ mice by bacterial 16S rRNA marker gene profiling on fecal samples from

*Rag1*HET and *Rag1*⁻/⁻ mice prior to antibiotic treatment, at the day of infection, and following establishment of persistent infection (day 36 post-infection [p.i.]). The null hypothesis that there was no difference in the microbial community of *Rag1*HET and *Rag1*⁻/⁻ mice was tested to determine if intrinsic differences in the microbiota could explain the differential outcome following FMT. Analysis of unweighted (Fig. 2a), weighted (Supplementary Fig. 3A) UniFrac or Bray-Curtis (Supplementary Fig. 3B) distances between samples did not delineate beta diversity differences in bacterial composition between *Rag1*HET and *Rag1*⁻/⁻ at day 36 p.i., despite significant shifts in microbiota composition of both groups following antibiotic treatment and subsequent *C. difficile* infection. Comparison of 16S rRNA bacterial community profiles between *Rag1*HET and *Rag1*⁻/⁻ mice by relative bacterial genera abundance (Fig. 2b and Supplementary Fig. 3c, d), UniFrac distances (Fig. 2c and Supplementary Fig. 3E–G) and unsupervised hierarchical clustering (Fig. 2d and Supplementary

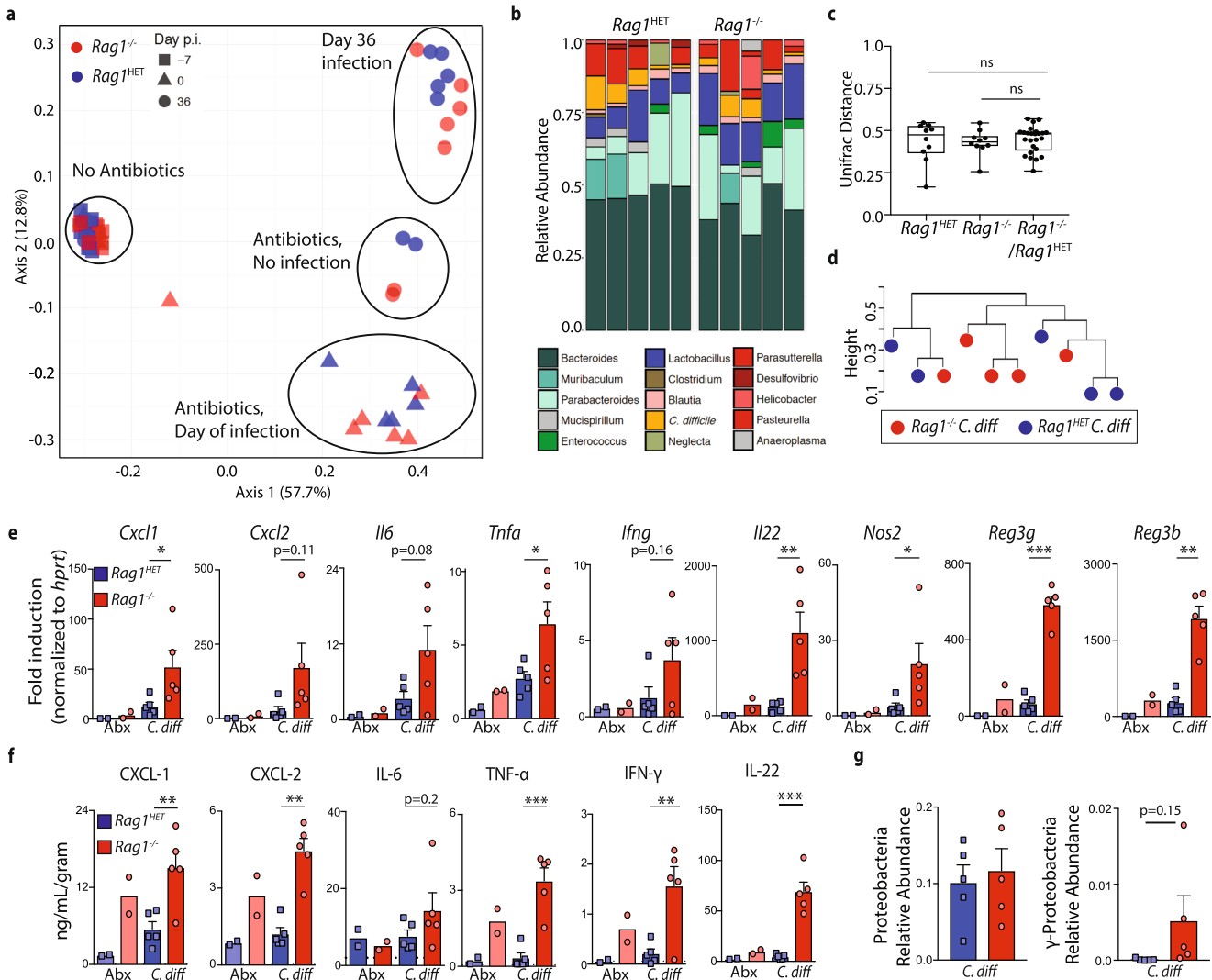

**Fig. 2 *C. difficile*-infected *Rag1*$^{-/-}$ and *Rag1*$^{HET}$ mice exhibit enhanced induction of proinflammatory immune response genes. a** Unweighted UniFrac principal coordinate analysis plot of 16S bacterial rRNA ASVs from fecal pellets of *Rag1*$^{-/-}$ and *Rag1*$^{HET}$ mice prior to ABX treatment, at day 0 of infection, and at day 36 post-*C. difficile* infection or ABX-treatment alone. Ellipses signify different experimental groups. **b** Relative abundance of top 15 bacterial ASVs from *C. difficile*-infected of *Rag1*$^{-/-}$ and *Rag1*$^{HET}$ mice at day 36 p.i. Bar plot is displayed at the genus level except for orange bars that represent an ASV aligning to *C. difficile*. **c** Unweighted UniFrac distance comparing the microbiota beta diversity dissimilarity within and between *C. difficile*-infected *Rag1*$^{-/-}$ and *Rag1*$^{HET}$ groups at day 36 p.i. One-way ANOVA conducted for statistical comparison. Boxes represent median, first and third quartile. Whiskers extend to the highest and lowest data point. **d** Dendrogram representation of intestinal microbial communities of *C. difficile*-infected of *Rag1*$^{-/-}$ and *Rag1*$^{HET}$ mice at day 36 p.i. using unsupervised hierarchical clustering of unweighted UniFrac distances to identify similarities between samples. **e** mRNA expression of proinflammatory immune response genes in whole colon tissue as assessed by qRT-PCR. Gene expression relative to ABX-treated *Rag1*$^{HET}$ mice and normalized to *Hprt*. Statistical significance was calculated by a two-sided unpaired *t*-test. *$p < 0.05$, **$p < 0.01$. Data are presented as mean values ± SEM. **f** Proinflammatory cytokine and chemokine protein levels in cecal tissue homogenates. Statistical significance was calculated by a two-sided unpaired *t*-test. *$p < 0.05$, **$p < 0.01$, ***$p < 0.001$. Data are presented as mean values ± SEM. **g** Relative abundance of Proteobacteria and gamma-Proteobacteria ASVs in fecal pellets of *Rag1*$^{-/-}$ and *Rag1*$^{HET}$ mice at day 36 p.i. Statistical significance was calculated by a two-sided unpaired *t* test. Data are presented as mean values ± SEM. Data presented in figure are representative of three independent experiments. Prior to ABX treatment (*n* = 10 *Rag1*$^{-/-}$; *n* = 12 *Rag1*$^{HET}$ mice), day 0 of infection (*n* = 6 *Rag1*$^{-/-}$; *n* = 5 *Rag1*$^{HET}$ mice), day 36 post-*C. difficile* (*n* = 5 *Rag1*$^{-/-}$ and *Rag1*$^{HET}$ mice) or ABX-treatment alone (*n* = 2 *Rag1*$^{-/-}$ and *Rag1*$^{HET}$ mice).

Fig. 3H) do not lead us to reject the null hypothesis that there was no microbial community level differences between groups. A PERMANOVA test of UniFrac distance indicated a statistically significant difference between the microbiota of *Rag1*$^{HET}$ and *Rag1*$^{-/-}$ mice prior to antibiotics as has been previously reported[43], but community differences were not statistically significant at later timepoints (Supplementary Table 1). Analysis of the 16S rRNA bacterial gene profile from a second validation cohort of *C. difficile* infected *Rag1*$^{HET}$ and *Rag1*$^{-/-}$ mice at time of FMT (day

32 p.i.) also failed to reject the null hypothesis as determined by unweighted UniFrac distances (Supplementary Fig. 4A), weighted UniFrac distances (Supplementary Fig. 4B), the magnitude of distances between groups (Supplementary Fig. 4C, E), unsupervised hierarchical clustering, and PERMANOVA tests of UniFrac distance between groups (Supplementary Fig. 4D, F and Supplementary Table 2).

16S rRNA marker gene profiling does not yield consistent species-level resolution of microbial communities[44], and therefore

may not reveal microbial differences that drive differential outcomes following FMT in *C. difficile*-infected *Rag1*[HET] and *Rag1*[−/−] mice. To address whether species-level differences at the time of FMT could drive FMT failure in *C. difficile* infected mice independent of the host's immune status, the cecal content of *C. difficile*-infected C57BL/6 or *Rag1*[−/−] mice was transferred into antibiotic-treated *Rag1*[−/−] or *Rag1*[HET] recipient mice, respectively (Supplementary Fig. 5A). Recipient mice established a *C. difficile* infection derived from the donor transplant, recovered from initial disease morbidity and then received a FMT from naïve C57BL/6 mice donors (Supplementary Fig. 5A). *Rag1*[HET] mice receiving the microbiota from a *C. difficile* infected *Rag1*[−/−] mouse were still capable of resolving *C. difficile* infection following FMT (Supplementary Fig. 5B–D). In contrast *Rag1*[−/−] mice receiving the microbiota from a *C. difficile* infected C57BL/6 mouse failed to clear *C. difficile* following FMT (Supplementary Fig. 5B–D). Next, germ-free (GF) C57BL/6 mice were cohoused with *Rag1*[−/−] or *Rag1*[HET] mice and subsequently treated with antibiotics, infected with *C. difficile* and then administered a FMT (Supplementary Fig. 5E). *C. difficile* infected ex-GF mice cohoused with either *Rag1*[−/−] or *Rag1*[HET] mice prior to infection were equally capable of resolving *C. difficile* infection (Supplementary Fig. 5F). This experimental result demonstrates that the microbiota from naive *Rag1*[−/−] mice does not inherently imprint FMT failure in immunocompetent mice. Combined, these data provide evidence that microbiota composition is not intrinsically sufficient to prevent *Rag1*[−/−] mice from resolving *C. difficile* infection following FMT.

Next, the intestinal inflammatory milieu of *Rag1*[HET] and *Rag1*[−/−] mice was assessed at time of FMT. *C. difficile*-infected *Rag1*[−/−] mice exhibited elevated expression of proinflammatory immune response genes in the colon (*Il22, Cxcl1, Cxcl2, Tnfa, Nos2, Reg3g*, and *Reg3b*) compared to *C. difficile*-infected *Rag1*[HET] mice (Fig. 2e). Lack of T and B cells in *Rag1*[−/−] mice can conflate interpretation of mRNA transcripts quantification from whole colon tissue. Therefore, total protein levels were measured to assess the in vivo concentration of these proinflammatory cytokines and chemokines regardless of contributing cellular composition. In agreement with gene transcriptional expression, *C. difficile*-infected *Rag1*[−/−] mice had increased protein levels of proinflammatory cytokines and chemokines in the large intestine at time of FMT compared to *C. difficile*-infected *Rag1*[HET] mice (Fig. 2f). Gastrointestinal infection-induced inflammation is associated with bystander γ-Proteobacteria bloom in the intestine[45,46]. Indeed 16S rRNA marker gene profiling revealed a trend, though not statistically significant, toward more γ-Proteobacteria in *C. difficile* infected *Rag1*[−/−] mice (Fig. 2g). These data support the hypothesis that FMT failure in *Rag1*[−/−] mice is driven by host immunity shaping intestinal environment at the time of FMT engraftment.

**CD4+ T cells support FMT-mediated resolution of *C. difficile* infection**. The inability of *Rag1*[−/−] mice to resolve *C. difficile* infection following FMT suggests adaptive immunity is important for FMT-mediated resolution of *C. difficile* infection. To determine the cellular components of the adaptive immune system that supports FMT, mice deficient in B cells (μMT[−/−] mice), CD8+ T cells (β₂M[−/−] mice) or CD4+ T cells (MHC Class-II[−/−] mice) were screened for FMT responsiveness following *C. difficile* infection in comparison to cohoused C57BL/6 mice. All four groups of mice exhibited similar morbidity and mortality following acute *C. difficile* infection and exhibited persistent *C. difficile* colonization (Supplementary Fig. 6A–C,[47]). Following FMT, C57BL/6, μMT[−/−], β₂M[−/−] mice resolved *C. difficile* infection (Fig. 3a) while MHC Class II[−/−] (C-II[−/−]) mice failed

to clear *C. difficile* (Fig. 3b) and maintained high toxin levels in the cecum (Fig. 3c). These data indicate that CD4+ T cells not B cells or CD8+ T cells are necessary for supporting FMT-mediated resolution of *C. difficile* infection.

Intestinal immune homeostasis is regulated, in part, by the balance of CD4+ T regulatory cells (T$_{reg}$) and T helper 17 (T$_H$17) cells responding and interacting with cues from the intestinal microenvironment[48,49]. *C. difficile* infection elicited a significant expansion in frequency (Fig. 3d) and total number of IL-17a+ competent (Fig. 3e) and IL-22+ competent (Fig. 3f) CD4+ T cells in the large intestine lamina propria. Further, ten days following FMT the T$_H$17 cell population remained significantly elevated relative to naïve mice despite resolution of *C. difficile* infection at this timepoint (Fig. 3d–f). To test the potential role of T$_H$17 cells in supporting FMT we genetically ablated the signature effector molecules for this T cell subset, IL-17a or IL-22, and assess FMT-mediated resolution following *C. difficile* infection. Both *Il17a*[−/−] (Supplementary Fig. 7A) and *Il22*[−/−] mice (Supplementary Fig. 7B) exhibited equivalent capacity to reduce *C. difficile* burden following FMT compared to littermate or cohoused wild-type mice. T-bet deficient (*Tbx21*[−/−]) mice that lack the capacity for T$_H$1 cell differentiation were also capable of resolving *C. difficile* infection following FMT (Supplementary Fig. 7C). These data suggest FMT-mediated resolution of *C. difficile* infection can occur independent of the T$_H$17 and T$_H$1 CD4+ T cell lineage.

**Depletion of CD4+ T-regulatory cells impairs FMT-mediated resolution of *C. difficile* infection**. In parallel with T$_H$17 cell expansion, the Foxp3+ and Foxp3+ Rorγt+ T$_{reg}$ cell significantly increased in the large intestine lamina propria following *C. difficile* infection and remained elevated following FMT compared to naive mice (Fig. 3g–i). To directly test the role of T$_{reg}$ cells in FMT-mediated resolution of *C. difficile*, the *Foxp3*[DTR] knockin mice system[50] was used to selectively deplete T$_{reg}$ cells in the *C. difficile* infected mice prior to FMT (Fig. 4a). Continued administration of diphtheria toxin (DT) drives systemic autoimmunity and mortality in these mice[50], therefore DT or PBS was administered only at day 8 and 9 post-infection, during the recovery from the acute phase of infection, to temporarily delete T$_{reg}$ cells. Diphtheria toxin administration resulted in a temporary 10–15% loss in weight in *C. difficile* infected mice (Fig. 4b), but minimal weight loss in antibiotic treated, uninfected mice (Supplementary Fig. 7D), suggesting the loss of T$_{reg}$ cells in combination with persistent *C. difficile* infection reactivated intestinal inflammation. DT treatment in *Foxp3*[DTR] mice did not alter *C. difficile* burden prior to FMT compared to PBS-treated *Foxp3*[DTR] mice (Fig. 4c). Following FMT, however, DT-treated *Foxp3*[DTR] mice exhibited a significantly delayed resolution of *C. difficile* infection compared to PBS-treated *Foxp3*[DTR] mice (Fig. 4c). Transient ablation of the large intestinal T$_{reg}$ cells by DT administration was confirmed at day 12 p.i. (Fig. 4d–f), and, in agreement with a previous report[51], the T$_{reg}$ cell population began to return by day 21 p.i. (day of FMT; Fig. 4d, g). Despite comparable total numbers, the relative proportion of T$_{reg}$ cells compared to T$_H$17 and T$_H$1 CD4+ T cell subsets remained diminished in DT-treated *Foxp3*[DTR] mice (Fig. 4h). The repopulated T$_{reg}$ cell compartment was not sufficient to limit intestinal inflammation as absolute numbers of T$_H$17, T$_H$1 CD4+ T cells, infiltrating inflammatory monocytes and neutrophils was increased in *C. difficile*-infected DT-treated *Foxp3*[DTR] mice compared to *C. difficile*-infected PBS-treated *Foxp3*[DTR] mice (Fig. 4i). Further, *C. difficile*-infected DT-treated *Foxp3*[DTR] mice had significantly elevated expression of proinflammatory immune defense genes in the colon at day 21 p.i. compared to *C. difficile*-infected PBS-treated *Foxp3*[DTR] mice (Fig. 4j), similar to the enhanced proinflammatory signature

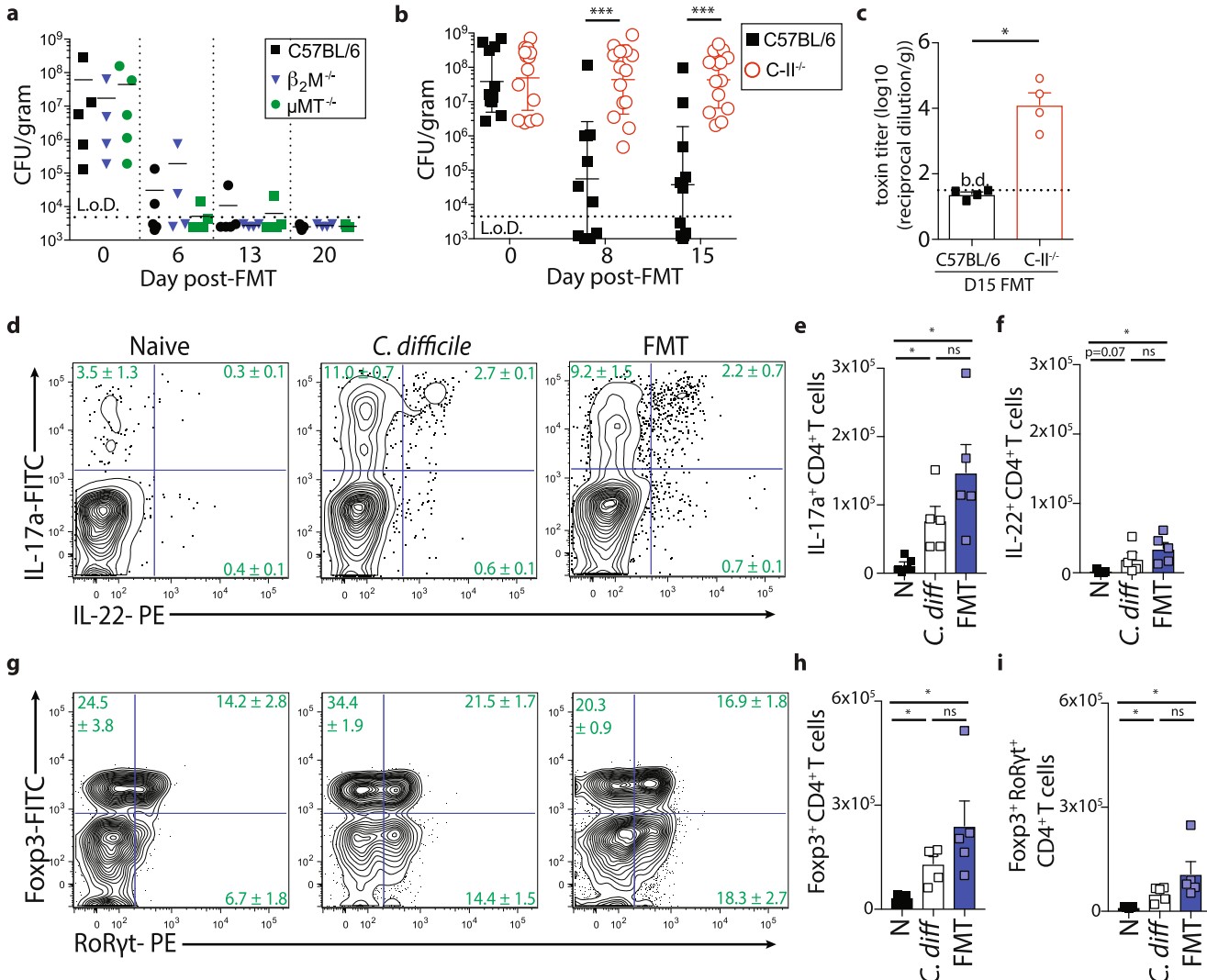

**Fig. 3 CD4⁺ T cells support FMT-mediated resolution of *C. difficile* infection. a** Cohoused C57BL/6, μMT⁻/⁻, β₂M⁻/⁻ mice were infected with *C. difficile* and bacterial burden was assessed in the fecal pellets following FMT. $n = 5$ C57BL/6; $n = 4$ β₂M⁻/⁻; $n = 5$ μMT⁻/⁻ mice. Data is representative of two independent experiments. **b** *C. difficile* burden in fecal pellets of cohoused C57BL/6 ($n = 11$) and C-II⁻/⁻ ($n = 14$) mice following FMT. Data is a combination of three independent experiments. Statistical significance was calculated by a two-sided unpaired *t*-test. ***$p < 0.0001$. Data are presented as mean values ± SEM. **c** Toxin titers in the cecal content at day 15 post-FMT. $n = 4$ mice per group. Statistical significance was calculated by two-sided Mann–Whitney test. *$p = 0.029$. Data are presented as mean values ± SEM. **d–f** Single cell suspensions isolated from the large intestine lamina propria of naïve, *C. difficile* infected, or *C. difficile*-infected FMT-treated C57BL/6 mice were stimulated ex vivo with PMA/Ionomycin in the presence of BFA and assessed for cytokine production. **d** Frequency and total number of **e** IL-17a or **f** IL-22 competent CD4⁺ T cells. FACS plots gated on live, CD45⁺, CD3ε⁺, CD4⁺ cells. Gating strategy shown in Supplemental Fig. 13. **g–i** Intranuclear transcription factor staining of cells isolated from the large intestine lamina propria of naïve, *C. difficile* infected, or *C. difficile*-infected FMT-treated C57BL6 mice. **g** Frequency and total number of **h** Foxp3⁺ or **i** Foxp3⁺ RoRγt⁺ CD4⁺ T cells. FACS plots gated on live, CD45⁺, CD3ε⁺, CD4⁺ cells. Data in **d–i** are representative of two independent experiments. Statistical significance was calculated by a two-sided unpaired *t*-test. $n = 5$ mice per group. **e** (N vs. *C. diff* *$p = 0.014$; N vs FMT *$p = 0.011$. **f**, N vs FMT *$p = 0.007$. **h**, N vs. *C. diff* *$p = 0.003$; N vs FMT *$p = 0.021$. **i** N vs. *C. diff* *$p = 0.004$; N vs FMT *$p = 0.028$) Data are presented as mean values ± SEM. *$p < 0.05$, **$p < 0.01$, ***$p < 0.001$. b.d. below detection.

observed in *C. difficile* infected *Rag1⁻/⁻* mice prior to FMT (Fig. 2e). These results demonstrate that depletion of the T_reg cells enhances expression of proinflammatory immune defense genes in the context of *C. difficile* infection and is sufficient to impair FMT-mediated resolution.

Continued administration of DT administration up through FMT also impaired resolution of *C. difficile* infection (Supplementary Fig. 7E), while DT treatment initiated immediately prior to FMT did not alter *C. difficile* resolution (Suppl. Fig 7F). These results indicate immune activation downstream of T_reg cell depletion is needed to re-shape the intestinal environment into a

state that is refractory to FMT. Combined, the data presented support a role for CD4⁺ T_reg cells in promoting the success of FMT therapy.

**FMT fails to engraft in *C. difficile* infected non-responsive mice.** To better understand the etiology of FMT failure, intestinal microbial community profiling was conducted following FMT in *C. difficile* infected *Rag1⁻/⁻* and C-II⁻/⁻ mice and compared to *Rag*^HET and C57BL/6 mice respectively. The microbiota of *C. difficile* infected FMT-treated *Rag1*^HET (Fig. 5a–d) or *C. difficile* infected FMT-treated C57BL/6 mice (Fig. 5e–h) shifted to

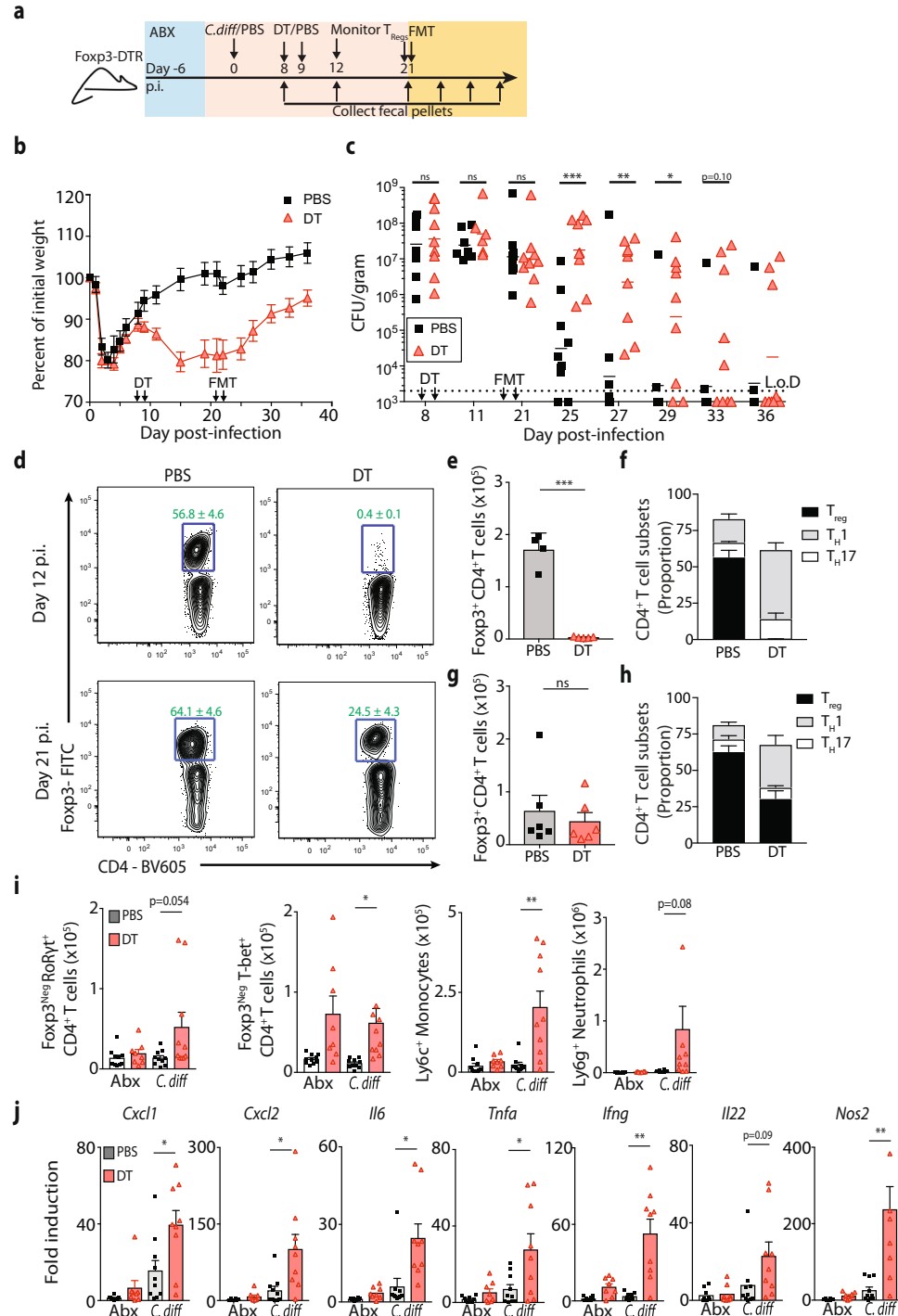

phenotypically resemble the composition of the FMT inoculum, therby demonstrating successful FMT engraftment. In contrast, the microbiota from *C. difficile* infected FMT-treated *Rag1*[−/−] (Fig. 5a–d) or *C. difficile* infected FMT-treated C-II[−/−] mice (Fig. 5e–h) remained distinct from the FMT inoculum. Analyses of weighted UniFrac (Supplementary Fig. 8a, e), or Bray-Curtis (Supplementary Fig. 8B, F) distances confirmed FMT engraftment in *Rag1*[HET] and C57BL/6 mice. In contrast, *C. difficile* infected FMT-treated *Rag1*[−/−] and C-II[−/−] mice failed to engraft the FMT inoculum. Analyses of the microbial community profile by relative bacterial genera abundance (Fig. 5b, f), unweighted or weighted UniFrac distance relative to the FMT (Fig. 5c, g and Supplementary Fig. 8C, G), unsupervised hierarchical clustering

(Fig. 5D, H and Supplementary Fig. 8d, h) and PERMANOVA test (Supplementary Table 3) all demonstrated the microbiota of FMT-responsive mice more closely resembled the FMT inoculum than the microbiota of FMT non-responsive mice. *C. difficile* infected *Rag1*[HET] or *Rag1*[−/−] mice receiving PBS instead of FMT did not exhibit significant microbiota composition changes (Supplementary Fig. 9A, B) and remained colonized with *C. difficile* (Supplementary Fig. 9C).

To identify the microbial species that differential engraft in FMT responsive vs. non-responsive mice, the 16S rRNA sequence dataset from *C. difficile* infected *Rag1*[HET] or *Rag1*[−/−] mice before and after FMT was analyzed for amplicon sequence variants (ASVs) that met the following criteria: (1) present in the FMT

**Fig. 4 T$_{reg}$ cell depletion impairs FMT-mediated resolution of *C. difficile* infection. a** Schematic of *C. difficile* infection, T$_{reg}$ cell depletion and FMT in *Foxp3*$^{DTR}$ mice. **b** Weight loss of *C. difficile* infected *Foxp3*$^{DTR}$ mice following DT ($n = 7$) or PBS ($n = 7$) administration. Data are presented as mean values ± SEM. **c** *C. difficile* burden in the feces of *Foxp3*$^{DTR}$ mice following DT or PBS administration and FMT. $n = 9$ mice per group. Data is a combination of three independent experiments. Statistical significance was calculated by a two-sided unpaired *t*-test. *$p = 0.015$, **$p = 0.001$, ***$p < 0.0001$. **d** Frequency of Foxp3$^+$ CD4$^+$ T cells in the large intestine lamina propria of *Foxp3*$^{DTR}$ mice at day 12 p.i. (day 4 post-DT treatment) or day 21 p.i. (day of FMT). $n = 4$–6 mice per group. FACS plots gated on live, CD45$^+$, CD3ε$^+$, CD4$^+$ cells. Gating strategy shown in Supplemental Fig. 13. **e** Total number of Foxp3$^+$ CD4$^+$ T cells and **f** relative proportion of CD4$^+$ T cell subsets in the large intestine lamina propria at day 12 p.i. (day 4 post-DT treatment). $n = 4$ PBS- treated *Foxp3*$^{DTR}$ mice; $n = 5$ DT-treated *Foxp3*$^{DTR}$ mice. Statistical significance was calculated by a two-sided Mann–Whitney test. ***$p < 0.0001$. Data are presented as mean values ± SEM. **g** Total number of Foxp3$^+$ CD4$^+$ T cells and **h** relative proportion of CD4$^+$ T cell subsets in the large intestine lamina propria at day 21 p.i. (day of FMT). $n = 6$ PBS & DT-treated *Foxp3*$^{DTR}$ mice. Statistical significance was calculated by a two-sided unpaired Mann-Whitney test. Data are presented as mean values ± SEM. **i** Total number of T$_H$17 cells, T$_H$1 cells, inflammatory monocytes, and neutrophils in the large intestine lamina propria at day 21 p.i. (day of FMT). $n = 10$ uninfected PBS-treated *Foxp3*$^{DTR}$ mice; $n = 8$ uninfected DT-treated *Foxp3*$^{DTR}$ mice; $n = 10$ *C. difficile* infected PBS-treated *Foxp3*$^{DTR}$ mice; $n = 10$ *C. difficile* infected DT-treated *Foxp3*$^{DTR}$ mice examined over three independent experiments. Statistical significance was calculated by two-sided unpaired *t*-test. *$p = 0.013$, **$p = 0.002$ Data are presented as mean values ± SEM. **j** mRNA expression of proinflammatory immune response genes in whole colon tissue as assessed by qRT-PCR. Gene expression relative to ABX-treated, uninfected *Foxp3*$^{DTR}$ mice and normalized to *Hprt*. $n = 10$ uninfected PBS-treated *Foxp3*$^{DTR}$ mice; $n = 8$ uninfected DT-treated *Foxp3*$^{DTR}$ mice; $n = 10$ *C. difficile* infected PBS-treated *Foxp3*$^{DTR}$ mice; $n = 9$ *C. difficile* infected DT-treated *Foxp3*$^{DTR}$ mice examined over three independent experiments. Statistical significance was calculated by two-sided unpaired *t*-test. *$p < 0.05$, **$p < 0.01$. Data are presented as mean values ± SEM.

inoculum, (2) absent in *C. difficile* infected *Rag1*$^{HET}$ or *C. difficile* infected *Rag1*$^{-/-}$ mice prior to FMT, (3) absent in *C. difficile* infected PBS-treated *Rag1*$^{HET}$ or *Rag1*$^{-/-}$ mice, (4) absent in *C. difficile* infected FMT-treated *Rag1*$^{-/-}$ mice, and (5) present in *C. difficile* infected FMT-treated *Rag1*$^{HET}$ mice. Despite these criteria, 279 ASVs met all five conditions with the majority of ASVs identified not distinguishable at the species levels (Supplementary Fig. 10).

FMT rejection required *C. difficile*-driven inflammation as ABX-treated, uninfected *Rag1*$^{-/-}$ mice (Supplementary Fig. 11A) successfully engrafted the FMT inoculum as determined by relative bacterial genera abundance (Supplementary Fig. 11B) unweighted (Fig. 11c), weighted (Supplementary Fig. 11D) UniFrac distances, beta diversity dissimilarity relative to the FMT (Supplementary Fig. 11E, G), and unsupervised hierarchical clustering (Supplementary Fig. 11F, H). Combined, these data demonstrate that the host's immune activation status can determine which transplanted bacteria successfully engraft in a *C. difficile*-infected host.

**FMT fails to restore intestinal metabolites in *C. difficile*-infected, FMT non-responsive mice.** Successful FMT treatment in recurrent *C.difficile* infected patients correlates with restoration of the intestinal metabolites to pre-infection concentrations[34,35]. Therefore, targeted metabolite profiling was conducted on cecal content from naïve, *C. difficile*-infected, and *C. difficile*-infected FMT-treated *Rag1*$^{Het}$ and *Rag1*$^{-/-}$ mice to assess whether FMT bacterial engraftment failure functionally impaired restoration of the intestinal metabolites. Amino acid[52,53], short chain fatty acid (SCFA)[54,55], primary and secondary bile acid[13,56–60] metabolites were selected based on reports demonstrating *C. difficile* colonization is influenced by the availability of these metabolites in the intestine. The metabolite composition in the cecum distinctly shifted between naïve and *C. difficile* infected mice (Fig. 6a, b), however, no significant difference was observed between *Rag1*$^{HET}$ and *Rag1*$^{-/-}$ mice prior to FMT (Fig. 6b and Supplementary Fig. 12A, B). Following FMT, the amino acids, SCFAs, 1° and 2° bile acids profile of *Rag1*$^{HET}$ mice but not *Rag1*$^{-/-}$ mice was restored to resemble the composition of naïve mice (Fig. 6a, b). These data demonstrate that failed engraftment of FMT bacterial species led to functionally impairment at the metabolite level. Following FMT, 2° bile acid derivatives were the most significantly enriched in *C. difficile* infected FMT-treated *Rag1*$^{HET}$ mice compared to *C. difficile* infected FMT-treated

*Rag1*$^{-/-}$ mice (Fig. 6c and Supplementary Fig. 12C). Specifically, deoxycholic acid, lithocholic acid, taurodeoxycholic acid, and omega-muricholic acid concentrations in the cecum of *C. difficile* infected FMT-treated *Rag1*$^{HET}$ mice were all restored to resemble the profile of naïve mice (Fig. 6d and Supplementary Fig. 8C). In contrast the 2° bile acid pool remained nearly undetectable in *C. difficile* infected FMT-treated *Rag1*$^{-/-}$ mice (Fig. 6d and Supplementary Fig. 12C) while the 1° bile acid pool remained elevated compared to naïve mice (Supplementary Fig. 12D, E). These data are in agreement with the observation that a canonical 2° bile acid converter bacterial specie, *Clostridium scindens*, engrafted in *C. difficile* infected FMT-treated *Rag1*$^{HET}$ mice but not *C. difficile* infected FMT-treated *Rag1*$^{-/-}$ mice (Supplementary Figs. 10 and 12F). Similarly, *C. difficile* infected FMT-treated C-II$^{-/-}$ mice exhibited significantly reduced cecal levels of deoxycholic acid and lithocholic acid compared to FMT-treated C57BL/6 mice (Supplementary Fig. 12G). These data identify intestinal metabolites composition, specifically 2° bile acids, as biomarkers for FMT engraftment failure in mice that exhibit elevated expression of immune defense genes. Combined, this report demonstrates that the immune status of the host is a critical factor in determining FMT therapy success in the context of *C. difficile* infection.

## Discussion

The clinical success of fecal transplants to treat recurrent *C. difficile* is an encouraging development for the treatment of this disease as well as other diseases associated with microbiota dysbiosis. The transplant's mechanism of action in *C. difficile* infection is postulated to be transplanted bacteria directly altering the intestinal environment to render it inhospitable for *C. difficile*. In this report, we find that successful engraftment of transplanted bacteria is dependent on the immune status of the recipient, specifically, immunodeficient hosts that lack CD4$^+$ Foxp3$^+$ T$_{reg}$ cells fail to resolved *C. difficile* infection following FMT. Persistent *C. difficile* infection in *Rag1*$^{-/-}$ mice or Foxp3$^+$ T$_{reg}$ cell-depleted mice led to exacerbated induction of inflammatory mediators in the large intestine compared to immunocompetent littermates. These FMT non-responsive mice exhibited impaired engraftment of the donor bacterial populations. Engraftment failure resulted in an inability to restore intestinal metabolite to pre-infection concentrations and ultimately failure to resolve *C. difficile* infection.

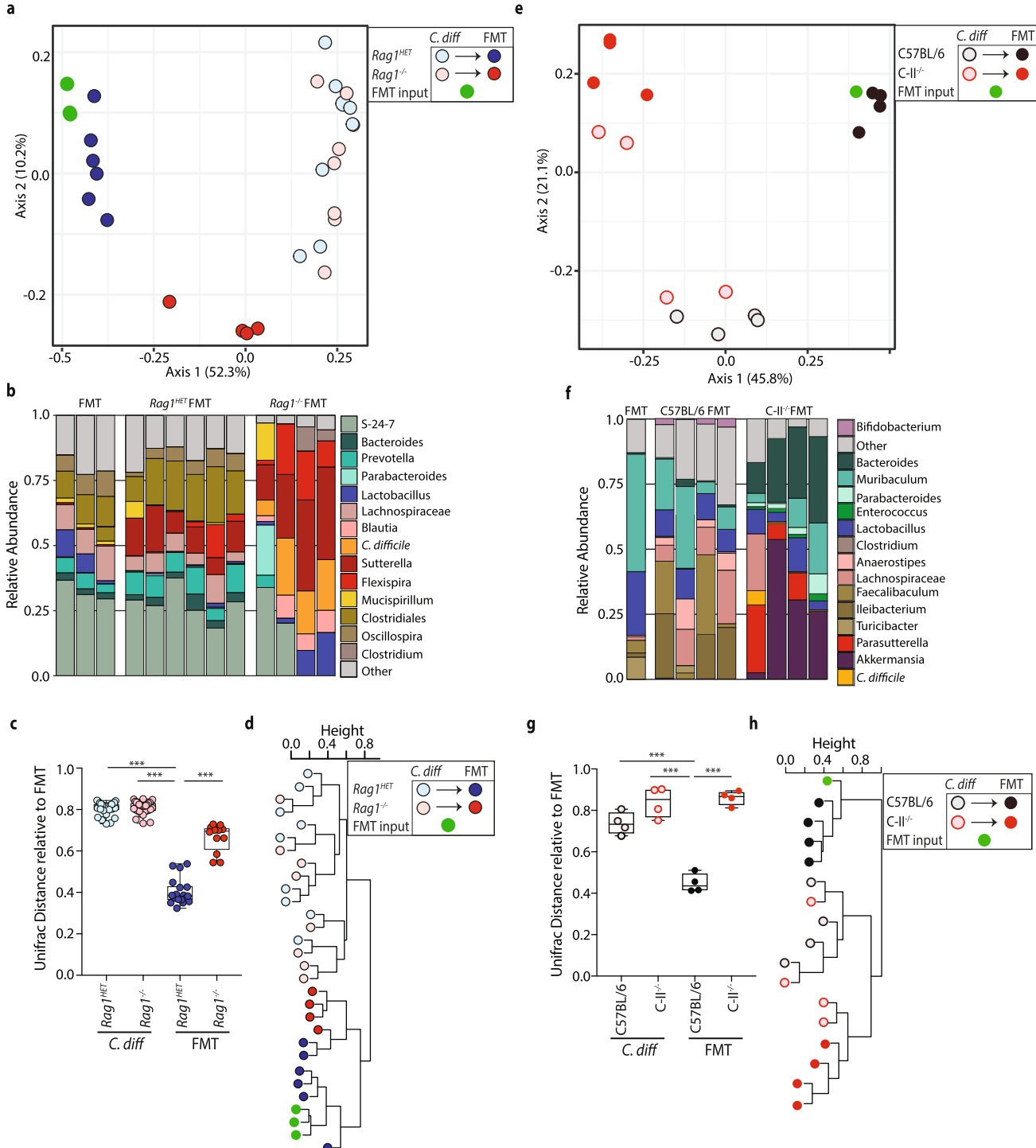

Foxp3$^+$ T$_{reg}$ cells serve an important balance in limiting host inflammation in the intestine. Unchecked, inflammatory mediators produced by the host may inhibit specific transplanted bacteria critical for the resolution of *C. difficile* infection. Host-derived inflammatory byproducts can be used by pathogens and bystander commensal species alike to promote their growth[61]. For example, Baumler and colleagues observed that host production of reactive oxygen and reactive nitrogen species induced by *Salmonella* infection could support expansion of both the pathogen and commensal *Enterococcus* species[62,63] while limiting the growth of commensal *Clostridia* species[64]. Further studies have demonstrated that ethanolamine, nitrates, formate, and

lactate metabolites induced by inflammation can support selective bacterial growth in the intestine[65–68]. Thus, a potential positive feedback loop exist by which the inflammatory environment induced by infection concurrently promotes the survival of select inflammation-tolerant bacterial species and inhibits engraftment of transplanted inflammation-sensitive bacteria. This feedback loop may be potentiated in the absence of functional immunoregulatory mechanisms, such as T$_{reg}$ cells that normally act as checkpoints that limit expansion of inflammation tolerant bacteria. For example, butyrate drives PPAR-γ signaling in epithelial cells and augments the intestinal T$_{reg}$ cell population ultimately depriving Enterobacteriaceae species of an oxygen electron

**Fig. 5 *C. difficile*-infected *Rag1*$^{-/-}$ and C-II$^{-/-}$ mice exhibit impaired FMT engraftment.** *C. difficile*-infected (**a–d**) *Rag1*$^{-/-}$ and *Rag1*$^{HET}$ mice or (**e–h**) C57BL/6 and C-II$^{-/-}$ mice were administered a FMT and microbial composition was analyzed. **a** Unweighted UniFrac principal coordinate analysis plot of 16S bacterial rRNA ASVs from the FMT inoculum, fecal pellets of *C. difficile* infected *Rag1*$^{-/-}$ and *Rag1*$^{HET}$ mice at the time of FMT, and cecal content of *Rag1*$^{-/-}$ and *Rag1*$^{HET}$ mice at day 21 post-FMT. **b** Relative abundance of top 15 bacterial ASVs in the microbiota of *C. difficile*-infected FMT-treated *Rag1*$^{-/-}$ and *Rag1*$^{HET}$ mice (day 21 post-FMT) Bar plot is displayed at the genus level except for orange bars that represent an ASV aligning to *C. difficile*. **c** Unweighted UniFrac distance comparing the microbiota beta diversity dissimilarity of *Rag1*$^{-/-}$ and *Rag1*$^{HET}$ groups to FMT inoculum. $n = 9$ *C. diff. Rag1*$^{HET}$; $n = 8$ *C. diff. Rag1*$^{-/-}$; $n = 6$ FMT *Rag1*$^{HET}$; $n = 4$ FMT *Rag1*$^{-/-}$ mice. Statistical significance was calculated by one-way ANOVA test using Dunnett method for multiple comparison adjustments. ***$p < 0.0001$. Boxes represent median, first and third quartile. Whiskers extend to the highest and lowest data point. **d** Dendrogram representation of intestinal microbial communities of FMT inoculum, *Rag1*$^{-/-}$ and *Rag1*$^{HET}$ groups using unsupervised hierarchical clustering of unweighted UniFrac distances to identify similarities between samples. **e** Unweighted UniFrac principal coordinate analysis plot of 16S bacterial rRNA ASVs from the FMT inoculum and fecal pellets from *C. difficile* infected C57BL/6 and C-II$^{-/-}$ mice at time of FMT and day 15 post-FMT. **f** Relative abundance of top 15 bacterial ASVs in the microbiota of *C. difficile*-infected FMT-treated C57BL/6 and C-II$^{-/-}$ mice (day 15 post-FMT). Bar plot is displayed at the genus level except for orange bars that represent an ASV aligning to *C. difficile*. **g** Unweighted UniFrac distance comparing the microbiota beta diversity dissimilarity of C57BL/6 and C-II$^{-/-}$ groups to FMT inoculum. $n = 4$ mice per group . Statistical significance was calculated by a one-way ANOVA test using Dunnett method for multiple comparison adjustments. ***$p < 0.0001$. Boxes represent median, first and third quartile. Whiskers extend to the highest and lowest data point. **h** Dendrogram representation of intestinal microbial communities of FMT inoculum, C57BL/6 and C-II$^{-/-}$ groups using unsupervised hierarchical clustering of unweighted UniFrac distances to identify similarities between samples. Data are presented as mean values ± SEM. ***$p < 0.001$.

acceptor[69]. The inflammatory byproducts produced in the intestine of FMT non-responsive hosts with defective immunoregulatory mechanisms may establish an environment that prevents the engraftment of inflammation-sensitive bacterial species. Indeed, in this report we observed 279 unique bacterial ASVs that successfully engrafted in *Rag1*$^{HET}$ mice but could not engraft in *Rag1*$^{-/-}$ mice, supporting this concept.

This report conducted a limited intestinal metabolite screen of amino acids, SCFA, 1° & 2° bile acids targeted to metabolites that have known roles in modulating *C. difficile* colonization in the intestine[13,52–60]. FMT non-responsive mice failed to restore 2° bile acid pools in the cecum. Primary and secondary bile acid levels are closely linked with *C. difficile* susceptibility[70,71]. Host-derived primary bile acids are germinants for *C. difficile* spores[72,73] while select commensal bacteria species capable of converting 1° bile acids to 2° bile acids can inhibit *C. difficile* growth[13,56–60]. Our results indicate 2° bile acids may serve as a useful biomarker for successful FMT engraftment and support a role for 2° bile acid restoration as one of the possible mechanisms that contribute to resolution of *C. difficile*. The metabolite screen performed in this study was limited in scope. A more extensive characterization of intestinal metabolome is needed to understand the relative contribution of bacterial-derived metabolites in FMT success and how they are shaped by the host's immune system.

Three recent reports observed that secondary bile acid derivatives can promote T$_{reg}$ cell development in the colon by directly signaling on T$_{reg}$ cells or indirectly signaling on dendritic cells that in turn promote T$_{reg}$ cell development[74–76]. Our data demonstrate low to absent 2° bile acid pools in *C. difficile* infected mice prior to FMT indicating that 2° bile acids are likely not promoting peripheral T$_{reg}$ cell development at the time FMT. Following FMT, however, restoration of 2° bile acids could promote T$_{reg}$ cell expansion and drive a feed forward loop that further reduces intestinal inflammation and enables repopulation of inflammation-sensitive commensal bacteria. FMT has previously been shown to promote the induction of anti-inflammatory properties via boosting IL-10 production in the context of a DSS-colitis mouse model[77]. The timing and context of this potential 2° bile acid - T$_{reg}$ cells feed forward mechanism in promoting bacterial engraftment will require further study.

This report assessed the ability of bacterial components of a FMT to engraft and resolve *C. difficile* infection in hosts with different immune statuses. The relative contribution of bacteriophages

transplanted in the FMT in resolving *C. difficile* infection was not assessed. A recent study found donor-derived bacteriophage populations to be associated with FMT success[78]. Bacteriophages may directly act on *C. difficile* itself through lytic processes[79,80] or stimulate immune cell activation in the intestine[81]. The relative contribution of bacteria and the bacteriophages in interacting with the immune system and supporting *C. difficile* resolution following FMT will require future exploration.

Use of FMT therapy in immunocompromised patient cohorts remains limited due to concerns about adverse effects[82]. Encouragingly, several case reports have described resolution of recurrent *C. difficile* in HIV$^{+}$, transplant-recipient, and cancer patients[83]. A multicenter, retrospective study found the FMT success rate in treating recurrent *C. difficile* in IBD patients on immunosuppressive therapy was comparable to the FMT success rates reported in IBD-free *C. difficile* patients though sustained engraftment of the transplanted bacteria was not assessed[84]. A subsequent study found the microbiota of immunocompromised hematopoietic stem cell recipients drifted away from the FMT inoculum over time[85] while recipients without overt immune defects exhibit long-term engraftment following FMT[86]. Combined, these reports suggest microbiota-based therapeutics for immunocompromised patients is possible, however, which immune pathways are dispensable will require further investigation and may vary depending on the disease being treated. For example, this report found that the immune deficiencies in the B-cell, CD8$^{+}$ T cell, T$_{H}$17 and T$_{H}$1 cell compartments did not impact FMT success while a defect in the T$_{reg}$ cell compartment impaired FMT-mediated resolution of *C. difficile* infection. Similar to immunosuppressive intervention strategies that reduce the risk of organ transplant rejection, identifying and targeting specific immunological pathways that dictate microbiota transplantation success or failure will be important for the development of more informed clinical approaches that can harness the microbiota to shape host physiology and alleviate disease.

## Methods

**Mice**. C57BL/6, *Rag1*$^{-/-}$, μMT$^{-/-}$, β$_2$M$^{-/-}$, C-II$^{-/-}$ (*H2*$^{dlAb1-Ea}$), *Il17a*$^{-/-}$, *Tbx21*$^{-/-}$, and *Foxp3*$^{DTR}$ mice were purchased from the Jackson Laboratory. *Il22*$^{-/-}$ mice were provided by R. Flavell (Yale University). All mouse strains were derived on a C57BL/6 background. All mice were bred and maintained in sterile autoclaved cages under specific pathogen-free conditions and kept on a grain-based diet (Labdiet 5053) at the Memorial Sloan Kettering Research Animal Resource Center or the University of Pennsylvania. Mice were provided autoclaved water ad libitum from water bottles. Sex and age-matched controls were used in all

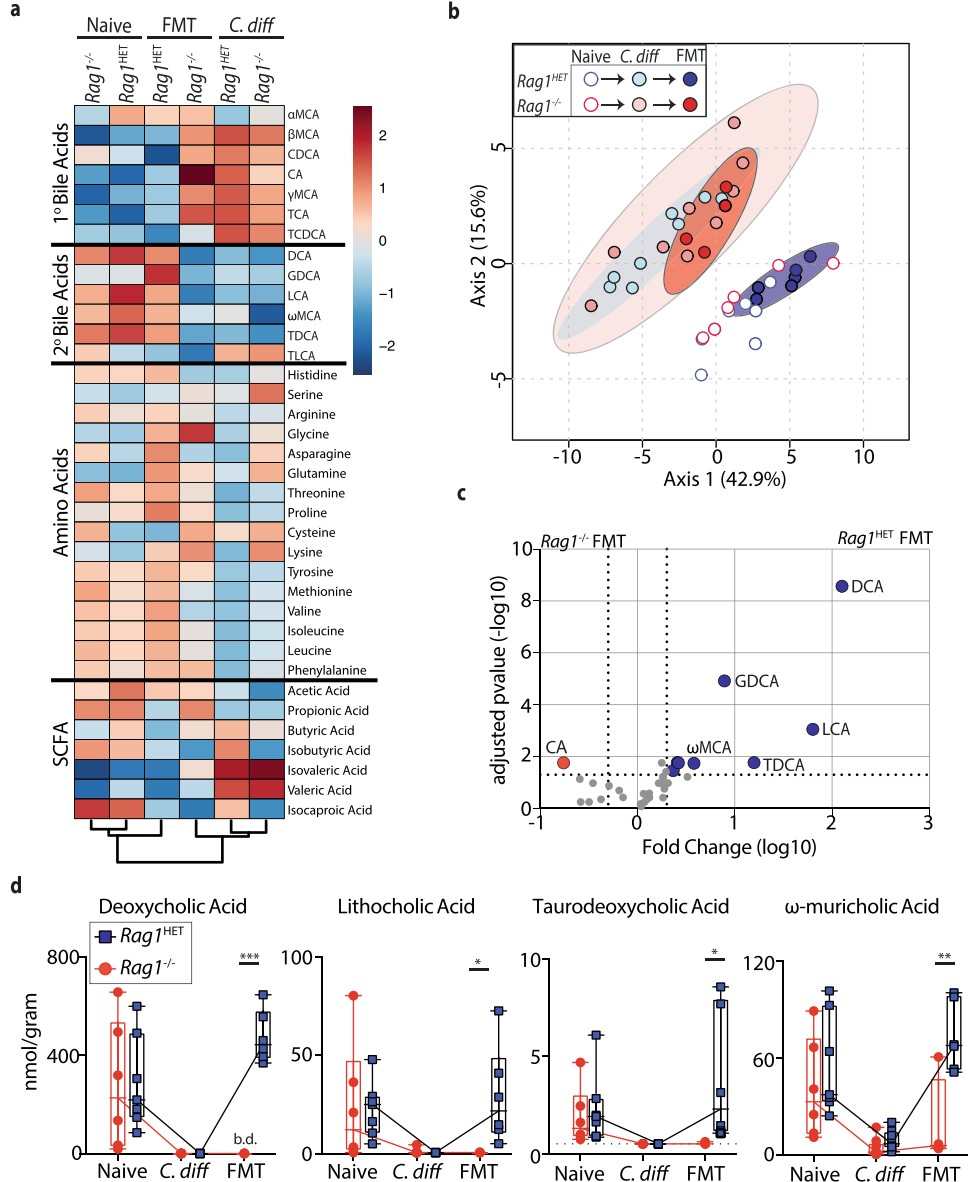

**Fig. 6 FMT non-responsive Rag1−/− mice fail to restore their intestinal metabolite profile to pre-infection conditions.** *C. difficile*-infected *Rag1*−/− and *Rag1*HET mice were administered FMT or PBS, sacrificed 21 days later along with naïve mice and amino acid, short chain fatty acid (SCFA), 1° and 2° bile acid pools were analyzed in cecal content. **a** Heatmap of relative concentration of amino acids, SCFA, 1° and 2° bile acids in cecal content of naïve (*n* = 5), *C. difficile*-infected (*n* = 6) or *C. difficile*-infected FMT-treated (*n* = 4 *Rag1*−/−; *n* = 6 *Rag1*HET mice). **b** Principal coordinate analysis plot comparing metabolite profile in the cecum of naïve, *C. difficile*-infected, and *C. difficile*-infected FMT-treated *Rag1*−/− and *Rag1*HET mice. Ellipses represent 95% confidence intervals. **c** Volcano plot of metabolites in the cecum of *C. difficile*-infected FMT-treated *Rag1*−/− and *Rag1*HET mice. Blue circles indicate metabolites enriched in *Rag1*HET mice. Red circles indicate metabolites enriched in *Rag1*−/− mice. Significance threshold criteria set at a two-fold change in concentration and an adjusted *p*-value of 0.05 using an unpaired *t*-test and adjusted for false discovery rate. **d** Concentration of individual 2° bile acids. *n* = 5 naïve *Rag1*−/− and *Rag1*HET mice; *n* = 6 *C. difficile*-infected *Rag1*−/− and *Rag1*HET mice; *n* = 4 *C. difficile*-infected FMT-treated *Rag1*−/−; *n* = 6 *C. difficile*-infected FMT-treated *Rag1*HET mice examined over two independent experiments. Statistical significance was calculated by a one-way ANOVA using Dunnett method for multiple comparison adjustments. DCA ***$p$ < 0.0001, LCA *$p$ = 0.043, TDCA *$p$ = 0.014, ωMCA **$p$ = 0.002. Boxes represent median, first and third quartile. Whiskers extend to the highest and lowest data point. *$p$ < 0.05, **$p$ < 0.01, ***$p$ < 0.001. TCDCA taurochenodeoxycholic acid, αMCA alphamuricholic acid, βMCA betamuricholic acid, γMCA gammamuricholic acid, CDCA chenodeoxycholic acid, TCA taurocholic acid, CA cholic acid, TLCA taurolithocholic acid, TDCA taurodeoxycholic acid, GDCA glycodeoxycholic acid, LCA lithocholic acid, ωMCA omegamuricholic acid, DCA deoxycholic acid. b.d. below detection.

experiments according to institutional guidelines for animal care. All animal procedures were approved by the Institutional Animal Care and Use Committee of the Memorial Sloan Kettering Cancer Center and University of Pennsylvania.

**Antibiotic pretreatment, *C. difficile* infection, and T_reg cell depletion.** Two to four month old mice were administered drinking water supplemented with neomycin (Sigma; 0.25 g/L), metronidazole (0.25 g/L), and vancomycin (Novaplus)

(0.25 g/L) for 72 h, and then replaced with normal water for the duration of the experiment. For experiments without littermates, mice were cohoused for three weeks prior to antibiotic treatment to allow for equilibration of the microbiota. Forty-eight hours following cessation of antibiotic water, mice received 200 μg of clindamycin (Sigma) by intraperitoneal injection. Twenty-four hours later, mice received ~1000 *C. difficile* spores (CD196, ribotype 027 strain[87], or VPI10463, ribotype 087 strain ATCC #43255) via oral gavage. After infection, mice were monitored for *C. difficile* burden in the feces and weight loss. *Foxp3*DTR mice were

administered diphtheria toxin (20 μg/kg of body weight) or PBS (i.p.) on consecutive days at timepoints indicated.

**Quantification of *C. difficile* burden and toxin.** Fecal pellets or cecal content were resuspended in deoxygenated PBS, and ten-fold serial dilutions were plated on BHI agar supplemented with yeast extract, taurocholate, L-cysteine, D-cycloserine, and cefoxitin at 37 °C in an anaerobic chamber (Coy Labs) overnight[88]. The presence of *C. difficile* toxins was determined using a cell-based cytotoxicity assay as previously described[89]. Briefly, human embryonic lung fibroblast WI-38 cells (ATCC# CCL-75) were incubated in a 96-well plate overnight at 37 °C. Ten-fold serial dilutions of supernatant from resuspended cecal content was added to WI-38 cells, incubated overnight at 37 °C and the presence of cell rounding was observed the next day. The presence of *C. difficile* toxin B was confirmed by neutralization by antitoxin antisera (Techlab, Blacksburg, VA). The data are expressed as the $\log_{10}$ reciprocal value of the last dilution where cell rounding was observed.

**FMT treatment.** The donor source for FMT was fecal pellets from naïve, C57BL/6 mice purchased from Charles River Laboratory and housed under specific pathogen-free conditions. Fecal pellets were collected fresh for each FMT and resuspended at a concentration of 0.2 g/mL of deoxygenated PBS under anaerobic conditions to preserve obligate anaerobe bacteria. FMT inoculum was flash spun to settle food debris and administered to mice via oral gavage (200 μl) and intrarectal instillation (100 μl) into chronically *C. difficile*-infected mice. Twenty-fours hours following initial FMT, mice receive a second dose of freshly prepared FMT via oral gavage (200 μl). To normalize FMT from experiment to experiment, fecal pellets were collected from the same colony of C57BL/6 mice. Following FMT, unless specified, mice were separated into individually housed cages.

**Bacterial DNA extraction, amplification, and analysis.** DNA was extracted from fecal pellet using a Qiagen Powersoil kit (Qiagen) according to manufacturer's instructions and 16S rRNA genes were amplified by PCR. Amplicons of the V4-V5 16S rRNA region were amplified and sequenced using an Illumina MiSeq platform as described in Buffie et al.[13]. Raw 16S sequence reads from experiments prior to FMT and with C-II$^{-/-}$ mice (Figs. 2 and 5e–h, and Supplementary Fig. 3 and 8E–H) were quality filtered and clustered into 97% ASVs using UPARSE[90] (USEARCH v11.0667) and ASV representatives were classified using BLAST against the NCBI refseq database as previously described[91]. Unweighted UniFrac distances were calculated using the phyloseq (v. 1.32.0)[92] R package using R version 3.2.1[93]. Relative abundance and PCoA plots were made using ggplot2 (v. 3.2.0)[94]. Weighted UniFrac distances were hierarchically clustered using the R function hclust from the stats package[93]. 16S rRNA sequencing data from experiments post-FMT and the validation cohorts (Fig. 5a–d, Supplementary Fig. 4, 8A–D, 9, and 11) was processed and analyzed using the QIIME2 pipeline[95]. DADA2[96], implemented as a QIIME2 plug-in, was used for sequence quality filtering. Taxonomic analysis was done using a Naïve Bayes classifier trained on the Greengenes 13_8 99% ASVs. For diversity metrics including UniFrac distances, a rooted phylogenetic tree was generated: first, a multiple sequence alignment was performed using MAFFT[97] and high variable positions were masked to reduce noise in a resulting phylogenetic tree. A mid-point rooted tree was then generated using FastTree[98].

**Tissue processing, RNA isolation, cDNA preparation, and qRT-PCR.** Cecal tissues were fixed with 4% paraformaldehyde, embedded in paraffin and 5 μm sections were cut and stained with hematoxylin and eosin. Crypt length was measured using ImageJ software (1.52q). Three distinct crypt lengths were measure for each sample and averaged. RNA was isolated from proximal colon tissue using mechanical homogenization and TRIzol isolation (Invitrogen) according to the manufacturer's instructions. cDNA was generated using QuantiTect reverse transcriptase (Qiagen). Quantitative RT-PCR was performed on cDNA using Taqman primers and probes in combination with Taqman PCR Master Mix (ABI). Genes of interest were displayed in arbitrary units relative to the expression of *Hprt*.

**Cytokine and chemokine quantification.** Cecal tissue was homogenized in 1 mL DMEM media for one minute using a stainless steel bead beater. Homogenates were centrifuged at $10,000 \times g$ for 5 min and supernatant collected. Supernatants were assessed for cytokine & chemokine concentrations using a Luminex bead array (Invitrogen) and concentrations displayed as μg of analyte per gram of cecal tissue.

**Isolation of cells from lamina propria, ex vivo cytokine detection, and flow cytometry.** Single cell suspensions were obtained from the large intestine lamina propria (Lp) by longitudinally cutting the cecum and colon then washing out content in PBS. Intestinal tissues were incubated at 37 °C under gentle agitation in stripping buffer (PBS, 5 mM EDTA, 1 mM dithiothreitol, 4% FCS, 10 μg/mL penicillin/streptomycin) for 10 min to remove epithelial cells followed by another 20 min to remove intraepithelial lymphocytes. The remaining tissue was digested with collagenase IV 1.5 mg/mL (500 U/mL), DNase 20 μg/mL in complete media (DMEM supplemented w/10% FBS, 10 μg/mL penicillin/streptomycin, 50 μg/mL gentamicin, 10 mM HEPES, 0.5 mM β-mercaptoethanol, 20 μg/mL L-glutamine) for 30 min at 37 °C under gentle agitation. Supernatants containing the Lp fraction

were passed through a 100-μm cell strainers and resuspended in 40% Percoll. Samples were then centrifuged for 20 min at $600 \times g$ to obtain Lp cell fractions. For direct ex vivo cytokine detection, cells isolated from the Lp were cultured in a 96-well plate in complete media and BFA (Golgiplug, eBioscience), Phorbol 12-myristate 13-acetate (50 ng/mL), and Ionomycin (500 ng/mL) for 4.5 h at 37 °C. Following incubation, cells were surface stained in FACS Buffer (PBS, 2% BSA, 0.2 mg Sodium Azide, 2 mM EDTA) using standard flow cytometric staining protocol with fluorescently conjugated antibodies specific to CD3ε (1-200 dilution), CD4 (1-200 dilution), CD5 (1-200 dilution), CD8α (1-200 dilution), CD11b(1-300 dilution), CD19 (1-100 dilution), CD45 (1-100 dilution), Ly6c (1-200 dilution), and Ly6g (1-100 dilution). After staining for surface antigens, cells were stained for intracellular cytokines using intracellular cytokine fixation buffer (eBioscience) and fluorescently conjugated antibodies specific for IL-22 (1-100 dilution), (clone 1H8PWSR, eBioscience), and IL-17a (1-300 dilution), (clone TC11-18H10.1, Biolegend). Cell viability was assessed with Live/Dead AQUA stain (Invitrogen). For intranuclear staining, the transcription factor staining kit (eBioscience) was used and cells stained with fluorescently conjugated antibodies specific for Foxp3 (1-300 dilution), (clone FJK-16S, eBioscience), RORγt (1-100 dilution), (clone B2D, eBioscience), T-bet (1-100 dilution), and (clone 4B10 eBioscience). Samples were collected by using an LSR II flow cytometer (Becton Dickinson). All flow cytometry data was analyzed by FlowJo v 9.9.6 (Treestar).

**Intestinal metabolite quantification.** Bile acids were quantified using a Waters Acquity uPLC System with a QDa single quadrupole mass detector[99,100]. Briefly, fecal samples were suspended in methanol (5 μL/mg stool), vortexed for 1 min, and centrifuged twice at $13,000 \times g$ for 5 min. The supernatants were analyzed on an Acquity uPLC with a Cortecs UPLC C-18 + 1.6 mm 2.1 × 50 mm column. The flow rate was 0.8 mL/min, the injection volume was 4 μL, the column temperature was 30 °C, the sample temperature was 4 °C, and the run time was 4 min per sample. Eluent A was 0.1% formic acid in water; eluent B was 0.1% formic acid in acetonitrile; the weak needle wash was 0.1% formic acid in water; the strong needle wash was 0.1% formic acid in acetonitrile. The seal wash was 10% acetonitrile in water. The gradient was 70% eluent A for 2.5 min, gradient to 100% eluent B for 0.6 min, and then 70% eluent A for 0.9 min. The mass detection channels were: +357.35 for chenodeoxycholic acid and deoxycholic acid; +359.25 for lithocholic acid; −407.5 for cholic, alphamuricholic, betamuricholic, gamma muricholic, and omegamuricholic acids; −432.5 for glycolithocholic acid; −448.5 for glycochenodeoxycholic and glycodeoxycholic acids; −464.5 for glycocholic acid; −482.5 for taurolithocholic acid; −498.5 for taurochenodeoxycholic and taurodeoxycholic acids; and −514.4 for taurocholic acid. Samples were quantified against standard curves of at least five points run in triplicate. Standard curves were run at the beginning and end of each metabolomics run. Quality control checks (blanks and standards) are run every eight samples.

Amino acids were quantified using a Waters Acquity uPLC System with an AccQ-Tag Ultra C18 1.7 μm 2.1 × 100 mm column and a Photodiode Detector Array[100,101]. Fecal samples were homogenized in methanol (5 μL/mg stool) and centrifuged twice at $13,000 \times g$ for 5 min. Amino acids in the supernatant were derivatized using the Waters AccQ-Tag Ultra Amino Acid Derivatization Kit (Waters Corporation, Milford, MA) and analyzed using the UPLC AAA H-Class Application Kit (Waters Corporation, Milford, MA) according to manufacturer's instructions. Quality control checks (blanks and standards) were run every eight samples. All chemicals and reagents used were mass spectrometry grade.

Short chain fatty acids were quantified using a Waters Acquity uPLC System with a HSS T3 1.8 μm 2.1 × 150 mm column and a Photodiode Detector Array[99,100]. Fecal samples were homogenized in volatile free fatty acid mix (5 μL/mg stool, AccuStandard, New Haven, CT) and centrifuged twice ($13,000 \times g$ for 5 min). The supernatant was filtered through 1.2, 0.65, and 0.22 μm filter plates (Millipore, Billerica, Massachusetts), and the filtrate was loaded into a total recovery vial (Waters, Milford, MA) for analysis. The flow rate was 0.25 mL/min, the injection volume was 5 μL, the column temperature was 40 °C, the sample temperature was 4 °C, and the run time was 25 min per sample. Eluent A was 100 mM sodium phosphate monobasic, pH 2.5; eluent B was methanol; the weak needle wash was 0.1% formic acid in water; the strong needle wash was 0.1% formic acid in acetonitrile, and; the seal wash was 10% acetonitrile in water. The gradient was 100% eluent A for 5 min, gradient to 70% eluent B from 5 to 22 min, and then 100% eluent A for 3 min. The photodiode array was set to read absorbance at 215 nm with 4.8 nm resolution. Samples were quantified against standard curves of at least five points run in triplicate. Standard curves were run at the beginning and end of each metabolomics run. Quality control checks (blanks and standards) were run every eight samples.

**Statistical analysis.** Results represent means ± SEM from distinct biological samples. For comparison of *C. difficile* burden, values were converted into $\log_{10}$ value and then statistical analysis was performed. Statistical significance was determined using a two-sided unpaired *t*-test for comparison between two groups. One-way ANOVA used for comparison between multiple groups. Statistical significance was determined using a Mann–Whitney test for small samples size ($n < 5$ per group) and with a non normal distribution. Statistical analyses were performed using Prism GraphPad software v6.0. (*$p < 0.05$; **$p < 0.01$; ***$p < 0.001$). For 16S sequencing data, in order to test the null hypothesis of no differences in the study group centroids, a PERMANOVA test was implemented by the function adonis() in the vegan package 2.5-5

[https://CRAN.R-project.org/package=vegan] was used in R 3.6.0. Metabolite profile was analyzed using Metabolanalyst. Log2 transformation and auto scaling of metabolite concentrations were performed to normalize the data. Significant metabolites were identified that met both a two-fold absolute value change threshold and 0.05 $p$-value using a $t$-test and adjusted for false discovery rate. Fold changes and adjusted $p$-values were log transformed and plotted on a volcano plot.

**Reporting summary**. Further information on research design is available in the Nature Research Reporting Summary linked to this article.

## Data availability

The authors declare that data supporting the findings of the study are available within the manuscripts, its supplementary information and the raw values are available in the source data file. Sequence data that support the findings of this study have been deposited in the NCBI SRA database. Accession number PRJNA668190. Source data are provided with this paper.

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

## Acknowledgements

We thank members of the Abt lab for helpful discussion and critical reading of the manuscript. We thank the Lucille Castori Center for Microbes, Inflammation and Center for assistance with high throughput sequencing. We thank Elliot Friedman of the PennCHOP Microbiome Program for conducting amino acid, SCFA and bile acid quantification. This research was supported by the NIH (R00 AI125786), the McCabe Fellowship Fund, and a PennCHOP Microbiome Pilot Grant.

## Author contributions

E.R.L. performed sequence data analysis, formatted figures, and edited manuscript. M.C.A. designed and executed the experiments, analyzed experimental data and wrote the manuscript. J.E.D, B.S., R.A.C., J.M., and I.Z. assisted in bacterial culturing, toxin assays, qRT-PCR, maintained and screened mouse colonies. M.S.S., R.M., and Z.A. performed experiments and data analysis with $Il17a^{-/-}$ or $Foxp3^{DTR}$ mice. B.F. and L.M.M. performed 16S rRNA high throughput sequencing. J.L. and K.B. contributed to 16S sequencing analysis and manuscript preparation.

## Competing interests

The authors declare no competing interests.

**Additional information**

