## [Peer Review File · Nature Communications]

Reviewers' comments:

Reviewer #1, expert in immune-microbial interactions in the intestine (Remarks to the Author):

In this manuscript, Littmann, et al. demonstrate that Rag^{-/-} mice do not properly engraft FMT and thus fail to clear *C. difficile* infection. These findings suggest an important role for the host immune system in enabling the engraftment of a donor microbiota. Unfortunately, the authors leave this finding as phenomenology and do not attempt to understand the mechanism by which the host immune system is important for engraftment. Rather, the paper focuses on multiple secondary measures of FMT engraftment, such as *C. difficile* clearance, degree of inflammatory response, and bile acids; however, it is not at all surprising that these elements—which are a direct consequence of FMT (e.g., a “normal” microbiota)—are not normalized in the setting of little to no engraftment of the donor stool.

Specific comments are as follows:

1. The observation that Rag^{-/-} mice do not properly engraft a FMT is quite interesting, but this work needs to be followed up to better understand the underlying mechanism (e.g., what specific B/T cell subtype is important, how do these cells influence engraftment). As an example, Sarkis Mazmanian's group demonstrated that IgA, which is deficient in Rag^{-/-} mice, is required for mucosal colonization of *B. fragilis*, with the idea that this may be a common theme. In the discussion, the authors highlight this avenue of dissecting immune pathways as a potential future direction but this additional characterization is particularly germane to the current paper and should be included here.
2. The authors demonstrate that Rag^{-/-} mice have more severe disease than WT mice (as assessed by multiple measures). Formally, it could be that FMT is less effective when disease is this severe, irrespective of the host immune status. Indeed, the authors even raise this as a possible confounding issue in the discussion but state that future studies will be needed to address it. In reality, this directly relates to the core finding of the manuscript and needs to be addressed in this manuscript. The authors should alter their model to establish WT mice that have more severe disease and/or lessen the severity of disease in the Rag^{-/-} mice to more conclusively demonstrate that it is the host genotype—not the severity of disease—which dictates whether the FMT will engraft.
3. It is not readily apparent that the *C. difficile* model used in this paper is particularly novel given that many other groups have done similar work and demonstrated that FMT is effective in infected mice (many of which are cited in the manuscript). As such, it seems that Figure 1 largely recapitulates findings from other papers and does not need to be a main Figure. If there is some aspect of this model that is innovative and/or new from previous reports, that point should be made more clear in the text.
4. Prior to experiments, the authors co-house WT and Rag^{-/-} mice for 3 weeks to “allow for equilibration of the microbiome.” Data should be provided to demonstrate that the microbiotas are indeed “equilibrated.” Based on their overall conclusion that donor stool does not engraft into a Rag^{-/-} mouse, it is not clear that the results of coprophagy will be any different than forced gavage. If the microbiotas of these mice do, in fact, “equilibrate,” then the host immune system is not critical for engraftment with a large enough (and/or multiply-dosed) inoculum.
5. The authors claim that FMT “is a distinct immunomodulatory event” (p5, line 21) given that it results in decreased expression of various inflammatory genes in *C. difficile*-infected mice. However, the FMT itself is not immunomodulatory (or at least no evidence is given of that); rather, it helps eradicate the infection, which then leads to decreased inflammation. In the same vein, one would not call antibiotics an immunomodulatory agent even though the findings would be similar.
6. The experiments in Suppl. Fig. 2 do not address their question of whether strain-specific differences might be driving the observed phenotypic differences. Given that the authors later demonstrate that Rag^{-/-} mice do not engraft a donor microbiota, the transfer of a WT microbiota into the Rag^{-/-} background similarly won't engraft. As such, these FMT-treated Rag^{-/-} mice still have what is largely a Rag^{-/-} microbiota, which is what the authors were trying to manipulate in this particular experiment. Instead of this approach, a cleaner experimental paradigm would be to

transfer WT or Rag-/- stool into GF B6 mice to tease out whether the endogenous microbiotas of these mice inherently lead to different disease and treatment manifestations. Similarly, one could transplant a WT microbiota into a GF Rag-/- mouse to more conclusively demonstrate that it is the host immune system and not differences in the microbiota that lead to the failed response to FMT. Both Penn and MSKCC have robust gnotobiotic facilities, so these experiments should be readily feasible.

7. On p8, line 12, the phrase "immune status" conjures up analysis of immune cell subsets rather than the authors' analysis of the intestinal inflammatory status as assessed by expression of inflammation-related genes. The authors should consider rephrasing.

8. In Fig 4A, there appears to be a shift in the FMT-treated Rag-/- mice albeit not a complete engraftment. Are there certain strains that are able to engraft in a Rag KO, suggesting that the adaptive immune system is only needed for certain species? More detailed microbiome analyses would be helpful.

9. It is not clear that the analyses related to *C. scindens* is necessary for this paper. As the authors allude to, this organism is one of many bacteria that is able to convert primary bile acids into secondary ones. The bile acid analyses are more relevant to this part of the story, though again, these differences are expected knowing that the donor stool doesn't engraft.

10. The authors highlight that there are several case reports of successful FMT in immunocompromised patients (in reality, there have been several hundred such patients who have been successfully treated with FMT). Regardless of the number of patients, these observations undercut the main conclusion of the paper that the host immune system is critical for engraftment (and therefore clinical resolution of *C. difficile* infection). Clarification should be provided.

Minor comments:

1. In the first sentence of the second paragraph of the introduction, it may be useful to clarify that *C. difficile* is the most common nosocomial infection encountered by hospitalized patients, not the most common infection not otherwise specified.

2. Metronidazole is no longer considered a first-line treatment option for *C. difficile* infections in adults (IDSA guidelines).

Reviewer #2, expert on immune control of intestinal microbiota (Remarks to the Author):

The authors established a model of persistent *C. diff* infection/inflammation that can be cured by FMT from normal mice. The authors then investigated differential FMT engraftment and *C. diff* clearance in wild-type vs Rag1-/- mice that lack adaptive T and B cells. The authors also investigated changes in bile acid metabolism and how these might impact on the observed differential clearance of *C. diff* following FMT.

While this is an interesting observation, I feel that, especially since the title mentions host immunity as main contributor, more careful investigation into what immunological mechanisms mediate the observed effect is necessary. While the section on bile acids is interesting it does not link back to host immunity at all.

Major comments:

The study is currently very descriptive with basically no cellular mechanism. Which adaptive cell types or other innate immune changes as a consequence of B and T cell deficiency contribute to the observed difference in fecal engraftment and clearance of *C. diff*? Is the same observed in T cell deficient (TCR β -/- or CD4-/-) or B cell deficient (JH-/-) mice? Can the phenotype be reserved by adoptively transferring a naive T cell pool from wild-type mice for example? Are ILC (or other innate immune cells) over-activated in the absence of T and B cells causing increased inflammation? Is increased inflammation in Rag1-/- due to missing regulation? Does transfer of Treg control increased *C. diff* inflammation in Rag1-/- mice?

While the section on bile acid metabolism is very interesting, it appears sudden and doesn't add to

what the title of the paper states. I would like to see a much more careful investigation into the lack of which (adaptive or innate) immune mechanisms cause the observed effect as suggested above.

Specific comments:

Figure 1:

- A) In the method section it is stated the mice receive 200 *C. diff* spores for infection but in this Figure 1A it is stated as 1000? Which one is correct?
- B) How does what is referred to in the text as 'initial disease morbidity' look like? What parameters have been measured and why is this data not shown?
- C) Why is the limit of detection (LOD) different in panel 1C (LOD=1) vs panel 2F (LOD=2)?
- D) I understand it is difficult to quantify immune cell infiltration and edema from these sections but is it possible to quantify/measure crypt elongation in these groups in order to statistically analyze them?

Figure 2:

- What was the rationale for removing the complete adaptive immune system? How does the adaptive immune system respond to FMT in wild-type mice? Is there for example a strong induction of Treg or IgA responses? This would provide some rationale for testing this in Rag1^{-/-} mice.
- It would be more logical to show panel C and D with the post infection first before the current panels A and B that are post FMT (so 100 days post infection).
- Has the microbiome composition been measured in this set of experiment to check for how much co-housing normalized the microbiome? I understand that the littermate experiments remedy this concern, but why not just show the littermate experiments? What does the co-housing experiment add to the story?
- Why was a different strain of *C. diff* used here. The use of two different strains throughout the manuscript should be better explained/justified. Are there any critical differences in virulence/pathogenicity between the two strains or can they be considered equal?
- No information is provided for the number of mice/group in the panels for Figure 2.

Figure 3:

- A) The Figure legend states n=3-5 mice per groups. However, group 'Antibiotics, no infection d36' seems to have only n=2 mice for each genotype in the PCoA plot. It would be interesting to also see the weighted UniFrac PCoA and/or Bray Curtis distance plots.
- B) It would be great to see analogous bar plots for the other time points as well in the supplementary Figure.
- D) Why is weighted unifracs used here while before unweighted was used?
- E&F) While it is clear that the Rag1^{-/-} mice have an elevated inflammatory status the authors should comment on the fact that the two groups of mice have a vastly different cellular composition of the colon tissue (Rag1^{Het} (WT) vs Rag1^{-/-} (no B and T cells), which makes normalizing and comparing expression levels difficult between the two groups.

Figure 4:

- A) It is known that an inflammatory milieu can promote blooming of certain bacteria such as Proteobacteria. Is something like this observed in the Rag1^{-/-} colony that has increased inflammatory markers?

Supplemental Figure 1:

- C) It is not clear to me what this PBS treatment control experiment controls for. See also Page 6, lines 8/9.

Reviewer 1:

Major Comments:

1. The Reviewer asks the authors to elaborate on the “quite interesting” observation that *Rag1*^{-/-} do not properly engraft a FMT by better understanding which specific B/T cell subtype drives the underlying immunologic mechanism.

We thank the Reviewer for this suggestion and agree this required significant more experimental investigation. We have made substantial progress in addressing this area and generated new experimental data since our first submission through a series experiments designed to dissect adaptive immune cellular compartments contributing to FMT success. This data is now described on in Fig. 3, Fig. 4, and Suppl. Fig. 5. The findings of these experiments are also summarized below:

- (i) We assessed the capacity of a FMT to resolve *C. difficile* infection in mice that lack B cells, CD8⁺ T cells or CD4⁺ T cells. This set of experiments described on Pg. 8, Ln. 14-25 and in Fig. 3A-C **reveal**

a necessary role for CD4⁺ T cells, but not B cells or CD8⁺ T cells, in resolution of *C. difficile* following FMT.

- (ii) Next, we characterized the CD4⁺ T cell response following *C. difficile* infection and FMT and found that both T_H17 and T_{Reg} cell subsets were expanded following *C. difficile* infection. These data are described on Pg. 9, Ln. 1-7, 16-18 and in Fig. 3D-I. We conducted FMT experiments with *C. difficile* infected IL17 and IL-22 deficient mice and observed no impaired in FMT-mediated resolution of *C. difficile* infection as described on Pg. 9, Ln 11-16 and in Suppl. Fig 5D-E. These data suggest T_H17 cell effector activity is not essential for FMT efficacy.
- (iii) To demonstrate the critical role for T_{Reg} cells in FMT-mediated clearance of *C. difficile* we undertook a series of experiments using *Foxp3*^{DTR} mice to transiently deplete T_{Reg} cells in *C. difficile* infected mice prior to FMT. Transiently ablation of T_{Reg} cell was necessary to prevent lethal, systemic autoimmunity from developing (Bos et al., 2013; Kim et al., 2007). T_{Reg} cell depletion resulted in impaired resolution of *C. difficile* following FMT demonstrating a critical role for this immune cell subtype. These data are described on Pg. 9-10, Ln. 14-26, 1-15 and in Fig. 4.
- (iv) To demonstrate that FMT-treated, CD4⁺ deficient (C-II^{-/-}) mice exhibit the same failure of FMT engraftment and reduced secondary bile acid pools as FMT-treated *Rag1*^{-/-} mice, we conducted parallel microbial community analysis and bile acid quantification in C-II^{-/-} mice following FMT. These analyses demonstrate FMT engraftment failure and diminished secondary bile acid levels in C-II^{-/-} mice. These data are described on Pg. 10-11, Ln. 17-26, 1-6 and in Fig. 5E-H, Fig. 6E-F, Suppl. Fig. 6E-H.

2. The Reviewer asks to alter the severity of disease in mice and to establish WT mice that have more severe disease and/or lessen the severity of disease in the *Rag1*^{-/-} mice to more conclusively demonstrate that it is the host genotype—not the severity of disease—which dictates whether the FMT will engraft.

We thank the Reviewer for this comment and agree that the severity of disease and the amount of inflammation induced by infection is an important factor in determining FMT success. We also tested our *C. difficile*/FMT model using a strain of *C. difficile* (VPI10463) that causes more severe disease in mice (Chen et al., 2008; Theriot et al., 2011)) and have included the results in Suppl. Fig. 2 D-E and described on Pg. 6, Ln. 13-20. FMT was successful in resolving VPI10463 infection in C57BL/6 mice but not *Rag1*^{-/-} mice.

We were also able to modify disease severity in wild-type mice but treating *Foxp3*^{DTR} mice with diphtheria toxin to deplete T_{Reg} cells. Prior to DT treatment *Foxp3*^{DTR} mice are immunologically indistinguishable from wild-type mice (Kim et al., 2007). Following *C. difficile* infection *Foxp3*^{DTR} mice exhibited a typical weight loss and recovery as displayed in Fig. 4E. Following DT treatment in *C. difficile* infected *Foxp3*^{DTR} mice we observed a temporary weight loss and increased expression of inflammatory genes as displayed in Fig. 4E,G and describe on Pg. 9-10, Ln 25-26 1-6. Loss of T_{Reg} cells resulted in delayed *C. difficile* resolution following FMT demonstrating that altering the immune status can alter severity of disease and FMT-mediated resolution of disease. In contrast, *Il22*^{-/-} mice experience severe disease following infection (Hasegawa et al., 2014) but are capable of resolving *C. difficile* following FMT (Suppl. Fig. 5E). Thus, we have been able to alter disease severity both by changing the strain of *C. difficile* and by selectively altering different immune compartments in the host. Increasing disease severity does not always impair FMT (e.g. VPI10463 infection, *Il22*^{-/-} mice). Instead manipulation of the host's immune status, specifically the T_{Reg} cell compartment (*Rag1*^{-/-}, C-II^{-/-}, *Foxp3*^{DTR}) determines the responsiveness to FMT.

3. The Reviewer asks to clarify what aspect of Figure 1 are innovative and/or new from previous reports.

We agree with the Reviewer that the primary purpose of Figure 1 in the original submission was to establish the *C. difficile*/FMT system. As such, we welcome the Reviewer's suggestion and have moved this figure to supplementary data (Suppl. Fig. 1). This figure demonstrates the resolution of intestinal inflammation by

histology, which we have quantified in new panel Suppl. Fig 1D. Further, the FMT indirectly shapes immune activation by reducing expression of proinflammatory genes following resolution of *C. difficile* infection. To our knowledge, neither of these points have been reported within the context of the *C. difficile*/FMT system and we have included text on Pg. 5, Ln 12-14 to emphasize these findings.

4. The Reviewer asks if the microbiome of WT and *Rag1*^{-/-} mice equilibrate after cohousing. And wonders if the microbiota of these mice do equilibrate than the host immune system may not be critical for engraftment.

We thank the Reviewer for bringing up this point and the opportunity to clarify our results. We assessed the microbiome of *Rag1*^{HET} vs. *Rag1*^{-/-} mice and C57BL/6 vs. C-11^{-/-} mice after cohousing and prior to *C. difficile* infection and did not observe any statistically significant differences in beta diversity in the microbial community composition. These results are displayed in Suppl. Fig. 3A-D, F and Reviewer Fig. 1.

Importantly, in this manuscript, we only report FMT engraftment failure in *Rag1*^{-/-} mice in the context of an ongoing *C. difficile* infection. *C. difficile* infection in *Rag1*^{-/-} or C-11^{-/-} mice induces intestinal inflammation and this intestinal environment does not support FMT engraftment and restoration of a healthy microbiome. Of note, in experiments describe in Suppl. Fig. 2B-E, mice were cohoused both prior and following FMT. *Rag1*^{-/-} mice still failed to resolve *C. difficile* infection despite prolonged exposure and presumed coprophagic ingestion of feces from wild-type cagemate mice that did successfully resolve infection following FMT. These points have been clarified on Pg. 6, Ln. 10-20.

5. The Reviewer ask the author to clarify the claim that the FMT “is a distinct immunomodulatory event”

We thank the Reviewer for the critique and agree this phrase does not adequately convey our message. The FMT indirectly shapes the immune status of the recipient by reducing expression of proinflammatory genes following resolution of *C. difficile* infection. We have modified Pg. 5 Ln 12-13 to read “FMT indirectly shapes the intestinal inflammatory environment via resolution *C. difficile* infection.

6. The Reviewer suggests transferring WT or *Rag1*^{-/-} stool into gnotobiotic mice as an alternative experimental approach.

We thank the Reviewer for the experimental suggestion. The experiment setup requires a microbiome transfer into gnotobiotic (GF) mice followed by subsequent antibiotic treatment to render the mice susceptible to *C. difficile* infection. Next, the antibiotic-treated, ex-GF mice would be infected with *C. difficile* and allowed to recover and then receive a FMT. Antibiotic treatment and subsequent *C. difficile* infection

shifts the ex-GF host's microbiome away from the original input inoculum (see Fig. 2 as an example). This microbial shift would complicate direct comparison of whether the original microbiome of naive WT vs *Rag1*^{-/-} mice was capable of supporting engraftment.

As an alternative approach, we attempted to use GF mice in the experimental design described in Supplemental Figure 4A. Unfortunately, inoculation of GF mice with cecal content from either *C. difficile* infected WT or *Rag1*^{-/-} resulted in a high acute mortality rate limiting our ability to perform subsequent FMT. The high mortality rate of GF mice from *C. difficile* infection has been previously observed (Reeves et al., 2012). With the remaining ex-GF mice we performed a FMT. We observed no difference in reduction of *C. difficile* burden between ex-GF mice reconstituted with a C57BL/6 microbiome and ex-GF mice reconstituted with a *Rag1*^{-/-} microbiome (Reviewer Fig. 2A,B).

Two caveats should be noted in this experiment. (1) We observed 1-2 log fold reduction of *C. difficile* burden in these ex-GF mice where as we observed complete resolution in ABX-treated mice in the experiment described in Supplemental Figure 4. (2) Logistically, the available gnotobiotic facilities do not have the capacity to dedicate experimental isolators or isocage system to this ~ 2 month long experiment. Therefore ex-GF mice were housed in autoclaved static cages following inoculation with cecal content. Due to small sample size and difficulty in getting enough ex-GF mice to survive initial infection we have not included this data in our resubmission. However, we are happy to include the data upon the Reviewer's request.

Reviewer Figure 2. GF B6 mice transplanted with microbiome derived from C57BL/6 or *Rag1*^{-/-} mice exhibit equivalent resolution of *C. difficile* following FMT. **(A)** Experimental schematic. Antibiotic-treated C57BL/6 and *Rag1*^{-/-} were infected with *C. difficile*. At day 30 p.i. cecal content (containing *C. difficile*) was transferred into reciprocal gnotobiotic (GF) C57BL/6 mice. At day 21 post cecal transplant, mice were administered FMT and **(B)** *C. difficile* burden monitored in the fecal pellets. Ex-GF B6 (B6 Microbiome) n=3. Ex-GF B6 (*Rag1*^{-/-} Microbiome) n=2.

7. The reviewer asks to rephrase “immune status” on pg 8, Ln 12 of the manuscript.

We agree with the Reviewer's comment and have changed this phrase to now state “inflammatory milieu” on Pg 7 Ln 23.

8. The reviewer asks for a more detailed microbiome analysis of mice following FMT, specifically what bacterial species do engraft in non-responsive *Rag1*^{-/-} mice.

We thank the Reviewer for the suggestion and the opportunity to improve the data in our manuscript. We conducted an analysis assessing how many ASVs met the following criteria:

1. Present in the FMT inoculum
2. Absent in *Rag1*^{HET} or *Rag1*^{-/-} prior to FMT
3. Absent in *Rag1*^{HET} or *Rag1*^{-/-} administered PBS
4. Present in *Rag1*^{HET} or *Rag1*^{-/-} administered FMT

318 ASVs met these criteria. Of these 318 ASVs, 39 were present in *Rag1*^{-/-} mice administered FMT. These data are described on Pg. 11, Ln. 10-16 and in Supplemental Figure 7. These data demonstrate that there is a subset of ASVs that do engraft in the *Rag1*^{-/-} mice.

9. The Reviewer ask for clarification as to why *C. scindens* analyses are included in the paper.

We agree with the Reviewer that this observation is secondary to bile acid quantification analyses displayed in the subsequent figure. Therefore, we have moved the *C. scindens* data to supplemental Figure 8A. *C. scindens* is a representative secondary bile acid producer. Identification of this ASV in the FMT inoculum, and FMT-treated *Rag1*^{HET} mice but not FMT-treated *Rag1*^{-/-} or PBS treated *Rag1*^{HET} mice is only suggestive of differences in the bile acid pools. We believe the secondary bile acid data is important to definitively demonstrate the link between failed FMT engraftment, inability to restore this key metabolite and unsuccessful resolution of *C. difficile* infection. As such, we have also now included primary and secondary bile acid quantification in C-II^{-/-} mice following FMT. Similar to *Rag1*^{-/-} mice, C-II^{-/-} mice exhibit significantly decreased secondary bile acid levels in the cecum following FMT compared to C57BL/6 mice. This data is described on Pg. 12, Ln. 10-13 and displayed in Fig. 6E,F, Suppl. Fig. 8I.

10. The Reviewer asks for clarification how the reports of immunocompromised patients successfully receiving FMT.

We thank the Reviewer for this comment and have expanded our discussion of this topic on Pg. 14-15 Ln 20-26, 1-8. Our data suggests that immune status is an important component of FMT success. Further, certain immunodeficiencies (B cell, CD8⁺ T cell, IL-22, IL17a deficient mice) do not impact FMT success while CD4⁺ T_{Reg} cell deficiency can impact FMT success. Our work highlights the nuanced nature of the immune system's role in FMT success. This complexity may require studies to distinguish the immunocompromised nature of transplant patients on immunosuppressive drugs versus HIV⁺ patients versus IBD patients. Further, a recent case study of an FMT-related death in an immunocompromised patient highlights the need for further research on this topic to better understand the immune system's role in FMT mechanism of action (DeFilipp et al., 2019).

Minor Comments

1. Clarify that *C. difficile* is the most common nosocomial infection encountered by hospitalized patients, not the most common infection not otherwise specified.

We thank the Reviewer for this comment and have adjusted the manuscript accordingly on Pg. 3, Ln 17-18 to state "most common nosocomial infection encountered by hospitalized patients".

2. Metronidazole is no longer considered a first-line treatment option for *C. difficile* infections in adults (IDSA guidelines).

We thank the Reviewer for the input and have removed Metronidazole from this sentence.

Reviewer 2:

General Comments:

1. The Reviewer asks the authors for a more careful investigation into what immunological mechanisms mediate the differential FMT engraftment and *C. difficile* clearance in wild-type vs. *Rag1*^{-/-} mice.

We thank the Reviewer for this suggestion and agree this required significant more experimental investigation. We have made substantial progress in addressing this area and generated new experimental data since our first submission through a series experiments designed to dissect adaptive immune cellular

compartments contributing to FMT success. This data is now described on in Fig. 3, Fig. 4, and Suppl. Fig. 5. The findings of these experiments are also summarized below:

- (i) We assessed the capacity of a FMT to resolve *C. difficile* infection in mice that lack B cells, CD8⁺ T cells or CD4⁺ T cells. This set of experiments described on Pg. 8, Ln. 14-25 and in Fig. 3A-C **reveal a necessary role for CD4⁺ T cells, but not B cells or CD8⁺ T cells, in resolution of *C. difficile* following FMT.**
- (ii) Next, we characterized the CD4⁺ T cell response following *C. difficile* infection and FMT and found that both T_H17 and T_{Reg} cell subsets were expanded following *C. difficile* infection. These data are described on Pg. 9, Ln. 1-7, 16-18 and in Fig. 3D-I. We conducted FMT experiments with *C. difficile* infected IL17 and IL-22 deficient mice and observed no impairment in FMT-mediated resolution of *C. difficile* infection as described on Pg. 9, Ln 11-16 and in Suppl. Fig 5D-E. These data suggest T_H17 cell effector activity is not essential for FMT efficacy.
- (iii) To demonstrate the critical role for T_{Reg} cells in FMT-mediated clearance of *C. difficile* we undertook a series of experiments using *Foxp3*^{DTR} mice to transiently deplete T_{Reg} cells in *C. difficile* infected mice prior to FMT. Transiently ablation of T_{Reg} cell was necessary to prevent lethal, systemic autoimmunity from developing (Bos et al., 2013; Kim et al., 2007). T_{Reg} cell depletion resulted in impaired resolution of *C. difficile* following FMT demonstrating a critical role for this immune cell subtype. These data are described on Pg. 9-10, Ln. 14-26, 1-15 and in Fig. 4.
- (iv) To demonstrate that FMT-treated, CD4⁺ deficient (C-II^{-/-}) mice exhibit the same failure of FMT engraftment and reduced secondary bile acid pools as FMT-treated *Rag1*^{-/-} mice, we conducted parallel microbial community analysis and bile acid quantification in C-II^{-/-} mice following FMT. These analyses demonstrate FMT engraftment failure and diminished secondary bile acid levels in C-II^{-/-} mice. These data are described on Pg. 10-11, Ln. 17-26, 1-6 and in Fig. 5E-H, Fig. 6E-F, Suppl. Fig. 6E-H.

Major Comments: The Reviewer asks if T cell or B cell deficient mice have the same phenotype as *Rag1*^{-/-} mice? Is increased inflammation in *Rag1*^{-/-} due to missing regulation and if the phenotype can be reversed by an adoptive naive T cell or specific Treg cell transfer?

In addition to the experimental results described above demonstrating that CD4 T cell deficient mice but not CD8 T cell or B cell deficient mice exhibited impaired FMT-mediated resolution of *C. difficile* infection, we conducted a series of experiments to attempt address whether the phenotype could be reversed by an adoptive T cell transfer.

We conducted an experiment transferring in bulk CD4⁺ T cell or sorted *Foxp3*^{GFP} T_{Reg} cells into a *C. difficile* infected *Rag1*^{-/-} mice. Twenty days following adoptive cell transfer, a FMT was administered and *C. difficile* burden was quantified. While littermate *Rag1*^{HET} mice resolved *C. difficile* infection following FMT, neither the *Rag1*^{-/-} mice receiving CD4⁺ T cells or T_{Reg} cells resolved infection (Reviewer Figure 3C). Characterization of the CD4⁺ T cell compartment in the large intestine lamina propria of recipient *Rag1*^{-/-} mice revealed the majority of transferred T cells no longer had T_{Reg} cell phenotype (Reviewer Fig 3A,B). Therefore, this experiment did not successfully restore the CD4⁺ T cell subset necessary to control inflammation and promote FMT engraftment. Transfer of CD4⁺ T cells into the lymphopenic environment of *Rag1*^{-/-} mice results in homeostatic proliferation to fill the empty T cell niche. These conditions promote homeostatic proliferation and cell differentiation of the transferred CD4⁺ T cells (Min et al., 2005; Stockinger et al., 2004). Spontaneous CD4⁺ T cell proliferation is dependent on the intestinal microbial composition and intestinal environment (Kieper et al., 2005). In the case of *C. difficile* infection, this environment is skewed to T_H17 cell differentiation and therefore prevents our ability to directly assess if a purified T_{Reg} cell infusion can regulate the intestinal environment of *C. difficile*-infected *Rag1*^{-/-} mice and promote FMT engraftment. We therefore employed a loss of function approach as an alternative and specifically ablated T_{Reg} cells using *Foxp3*^{DTR} mice as described on Pg. 9-10, Ln. 14-26, 1-15 and displayed in

Fig. 4.

Reviewer Figure 3. Adoptively transferred CD4⁺ T cells differentiate into T_H17 cells in *C. difficile* infected *Rag1*^{-/-} mice and FMT fails to resolve infection. (A-C) 2x10⁶ bulk CD4⁺ T cells or FACS-sorted GFP⁺ Foxp3⁺ T_{Reg} cells were transferred i.v. into *Rag1*^{-/-} mice 20 and 1 day prior to FMT. (A) Frequency and (B) total number of Rorgt⁺ and Foxp3⁺ T reg cells in the large intestine lamina propria at day 20 post-FMT. (C) *C. difficile* burden in the feces following CD4⁺ T cell transfer and subsequent FMT.

Specific Comments:

Figure 1:

A) In the method section it is stated the mice receive 200 *C. diff* spores for infection but in this Figure 1A it is stated as 1000? Which one is correct?

We thank the Reviewer for identifying this error. Our *C. difficile* infection system administers approximately 1,000 spores. We have corrected the method section on Pg. 16, Ln 12 to reflect this.

B) How is initial disease morbidity measured?

We agree that disease morbidity parameters should be described and displayed for the benefit of the readers. Disease was measured by weight loss and are now described in the Methods section on Pg. 16, Ln. 14 and data is display in Figure 1B. We observed no difference between *Rag1*^{HET} and *Rag1*^{-/-} mice as previously reported by (Abt et al., 2015; Hasegawa et al., 2014).

C) Why is the limit of detection (LOD) different in panel 1C (LOD=1) vs panel 2F (LOD=2)?

We thank the Reviewer for pointing out this difference. The limit of detection of our cell based toxin titer assay changed based on the amount of input material available. In panel 1C (now Suppl. Fig. 1B) the assay was done on supernatant from cecal content, in panel 2F (now Fig. 1D) the toxin assay was done on fecal supernatants. Since there was less total grams of starting material with the fecal supernatants, the limit of detection in this experiment was less sensitive. We thank the Reviewer for this comment. We re-check our limit of detection for these two panel and indeed found that panel 1C L.o.D. was actually 1.5. This has been corrected in Suppl. Fig 1B.

D) Is it possible to quantify/measure crypt elongation in these groups in order to statistically analyze them?

We thank the Reviewer for this suggestion and we have measured the average crypt length for each group. FMT significantly reduces the crypt length in C57BL/6 mice but fails to limit crypt elongation in *Rag1*^{-/-} mice. This data are now displayed in Suppl. Fig. 1D and Fig. 1F and analysis method described on Pg. 18, Ln 14-15.

Figure 2:

A) What was the rationale for removing the complete adaptive immune system?

We thank the Reviewer for raising this point and the opportunity to address this rationale. Both wild-type and *Rag1*^{-/-} mice are capable of recovering from initial disease however both mice remained colonized with *C. difficile*, suggesting the immune system is not an important factor in pathogen clearance ((Abt et al., 2015; Hasegawa et al., 2014) Fig. 1B, Suppl. Fig. 1A). Further, Leslie et al, observed the antibiotic pretreatment regimen and starting microbiome composition, not presence of T and B cells, determined whether *C. difficile* infection persisted (Leslie et al., 2019). Combined, these data suggest FMT therapy acts independently of an intact adaptive immune system to resolve *C. difficile*. Therefore, we decided to test this null hypothesis that FMT would be equally successful regardless of host immune status. To our surprise we observed that persistently infected *Rag1*^{-/-} mice were not capable of resolving infection following FMT, thereby refuting our null hypothesis. This rationale is now described on Pg. 5-6 Ln 17-26, 1-4.

B) How does the adaptive immune system respond to FMT in wild-type mice?

Specific bacteria population can shape CD4⁺ T cell population (e.g. SFB promote T_H17 cell; *B. fragilis* or *Clostridia* promote T_{Reg} cells). Further FMT therapy has been demonstrated to promote IL-10⁺ CD4⁺ T cells (Burrello et al., 2018). Within the context of our *C. difficile*/FMT system we have generated new experimental data assessing the CD4⁺ T cell response following *C. difficile* infection and subsequent FMT. *C. difficile* infection elicited a significant expansion in frequency and total number of IL-17a⁺ competent and IL-22⁺ competent CD4⁺ T cells in the large intestine lamina propria (Fig 3D-F). Ten days following FMT these T_H17 cell population remained significantly elevated relative to naïve mice despite resolution of *C. difficile* infection at this timepoint (Fig. 3D-F). In parallel with T_H17 cell expansion, the Foxp3⁺ T_{Reg} cell population significantly increased in the large intestine lamina propria following *C. difficile* infection and remained elevated following FMT compared to naïve mice (Fig. 3G-I) These data are described on Pg. 9, Ln. 1-7, 16-18 and displayed in Fig. 3D-I.

C) It would be more logical to show panel C and D with the post infection first before the current panels A and B that are post FMT.

We thank the Reviewer for this suggestion to improve the flow of the manuscript. We have rearranged this figure (now figure 1). The figure now starts with data post infection and continues with data post FMT.

D) Why not just show the littermate experiments? What does the cohousing experiment add to the story?

We believe this is a reasonable set of questions by the Reviewer. We agree that the cohousing experiments are complementary to the littermate experiments. As such, we have moved these experiments to supplemental figures. The cohousing experiments displayed in supplemental figure 2 demonstrate two key points:

1). Cohousing experiments described in supplemental Figure 2 assessed whether prolonged exposure to a 'healthy' microbial environment (via coprophagy) can impact *C. difficile* infection resolution in *Rag1*^{-/-} mice. (see Response to Reviewer 1: Point 4).

2.) New experimental data added in our resubmission relies on the cohousing of wild-type and knockout strains due to logistical breeding constraints. Therefore, demonstrating that cohoused C57BL/6 and *Rag1*^{-/-} mice exhibit the same phenotype as littermate experiments is an important piece of data that supports subsequent cohousing experiments. Normalization of the microbiome via cohousing has been reported in the literature supporting this line of experimentation (Ubeda et al., 2012).

E) Why was a different strain of *C. diff* used here. The use of two different strains throughout the manuscript should be better explained/justified?

We thank the Reviewer for the question and agree that the use of the two different strains requires further explanation. The CD196 strain is a member of the ribotype 027 group, which is prevalent cause of hospital outbreaks (Stabler et al., 2010). CD196, however, has a lower pathogenic profile (Jarchum et al., 2012), therefore we decided to also test the highly virulent VPI10463 strain (Chen et al., 2008; Theriot et al., 2011) to increase the disease severity in both wild-type and *Rag1*^{-/-} mice and assess whether the differential response to FMT persisted (see Response to Reviewer 1: Point 2). We have included text on Pg. 6, Ln. 13-20 to provide the reader with context for the use of the two strains of *C. difficile*.

F) No information is provided for the number of mice/group in the panels for Figure 2.

We thank the Reviewer for identifying this deficiency. We have now included the number of mice/group in the figure legend of all figures.

Figure 3:

A) The Figure legend states n=3-5 mice per groups. However, group ‘Antibiotics, no infection d36’ seems to have only n=2 mice for each genotype in the PCoA plot. It would be interesting to also see the weighted UniFrac PCoA and/or Bray Curtis distance plots.

We thank the Reviewer for identifying this discrepancy. We have adjusted the figure legend to read: “Unweighted UniFrac principal coordinate analysis plot of 16S bacterial rRNA OTUs from fecal pellets of *Rag1*^{-/-} and *Rag1*^{HET} mice prior to ABX treatment n=10-12, day 0 of infection (n=8), or day 36 post- *C. difficile* (n=5) or mock infection (n=2)” on Pg. 21-22, Ln 26,1-3.

B) Also include the weighted UniFrac PCoA and/or Bray Curtis distance plots in addition to unweighted PCoA.

We agree with the Reviewer that weighted UniFrac PCoA and Bray Curtis distance plots strengthens these data. Therefore, we have included both kinds of plots for microbial community profiling done prior to FMT and following FMT. These data are now displayed in Suppl. Fig. 3A-B, Suppl. Fig 6A,B, Suppl. Fig. 6E,F.

C) Include analogous bar plots for the other time points as well in the supplementary Figure.

We agree with the Reviewer that abundance plots at other time points strengthen these data. Therefore, we have included abundance plots for pre-antibiotics and day of infection in Suppl. Fig 3C,D). We have also included abundance plots for C-II^{-/-} vs. C57BL/6 mice in Fig. 5F.

D) Why is weighted unifrac used here while before unweighted was used?

Our goal was to provide both unweighted (via PCoA) and weighted (via dendrogram) measurements for the reader. However, we agree with the Reviewer that is method to display the data is confusing. Therefore we have rearranged the figures so that all microbial profiling data in the main figure is derived from the unweighted UniFrac distance and microbial profiling data in the supplemental figures are derived from weighted UniFrac distance (with the exception of labeled Bray-Curtis plots). We believe this presentation of the data provides the reader with a clear and comprehensive appreciation of the data and we thank the Reviewer for this comment.

E) The authors should comment on *Rag1*^{HET} and *Rag1*^{-/-} mice having different cellular composition in the colon making comparison of gene expression levels difficult

We agree with the Reviewer and thank the reviewer for this comment. We have added text on Pg 8 Ln 2-5 stating: “Lack of T and B cells in *Rag1*^{-/-} mice conflate interpretation of gene expression quantification from whole colon tissue therefore total protein levels were measured to assess the *in*

vivo concentration of these proinflammatory cytokines and chemokines regardless of contributing cellular composition.”

Figure 4:

A) It is known that an inflammatory milieu can promote blooming of certain bacteria such as Proteobacteria. Is something like this observed in the Rag1^{-/-} colony that has increased inflammatory markers?

We thank the Reviewer for this suggestion and we have included data assessing the abundance of Proteobacteria and gamma-Proteobacteria in *C. difficile* infected Rag1^{HET} and Rag1^{-/-} mice. We observed a trend toward more gamma-Proteobacteria in *C. difficile* infected Rag1^{-/-} mice compared to *C. difficile* infected Rag1^{HET} mice. This trend was observed in two independent experiments however, it did not reach statistical significance perhaps indicating this readout for intestinal inflammation may not be sensitive enough in this experimental system. This data is described on Pg 8 Ln 7-10 and display in Fig. 2G. Further, we conducted this analysis in new microbial profiling data between *C. difficile* C57BL/6 and C-Il^{-/-} mice and observed a similar pattern (Reviewer Fig. 4).

Supplemental Figure 1:

A) It is not clear what this PBS treatment control experiment controls for.

We thank the Reviewer for the opportunity to clarify our experimental design. This control demonstrates C57BL/6 and Rag1^{-/-} mice are stably infected with *C. difficile* and PBS instillation is not sufficient to resolve infection. However, this piece of data is redundant with data displayed in Suppl. Fig. 1A and we have therefore removed it from this version of the manuscript.

Abt, M.C., Lewis, B.B., Caballero, S., Xiong, H., Carter, R.A., Susac, B., Ling, L., Leiner, I., and Pamer, E.G. (2015). Innate Immune Defenses Mediated by Two ILC Subsets Are Critical for Protection against Acute Clostridium difficile Infection. *Cell Host Microbe* 18, 27-37.

Bos, P.D., Plitas, G., Rudra, D., Lee, S.Y., and Rudensky, A.Y. (2013). Transient regulatory T cell ablation deters oncogene-driven breast cancer and enhances radiotherapy. *J Exp Med* 210, 2435-2466.

Burrello, C., Garavaglia, F., Cribsiu, F.M., Ercoli, G., Lopez, G., Troisi, J., Colucci, A., Guglietta, S., Carloni, S., Guglielmetti, S., *et al.* (2018). Therapeutic faecal microbiota transplantation controls intestinal inflammation through IL10 secretion by immune cells. *Nat Commun* 9, 5184.

Chen, X., Katchar, K., Goldsmith, J.D., Nanthakumar, N., Cheknis, A., Gerding, D.N., and Kelly, C.P. (2008). A mouse model of Clostridium difficile-associated disease. *Gastroenterology* 135, 1984-1992.

DeFilipp, Z., Bloom, P.P., Torres Soto, M., Mansour, M.K., Sater, M.R.A., Huntley, M.H., Turbett, S., Chung, R.T., Chen, Y.B., and Hohmann, E.L. (2019). Drug-Resistant E. coli Bacteremia Transmitted by Fecal Microbiota Transplant. *The New England journal of medicine* 381, 2043-2050.

- Hasegawa, M., Yada, S., Liu, M.Z., Kamada, N., Munoz-Planillo, R., Do, N., Nunez, G., and Inohara, N. (2014). Interleukin-22 regulates the complement system to promote resistance against pathobionts after pathogen-induced intestinal damage. *Immunity* *41*, 620-632.
- Jarchum, I., Liu, M., Shi, C., Equinda, M., and Pamer, E.G. (2012). Critical role for MyD88-mediated neutrophil recruitment during *Clostridium difficile* colitis. *Infect Immun* *80*, 2989-2996.
- Kieper, W.C., Troy, A., Burghardt, J.T., Ramsey, C., Lee, J.Y., Jiang, H.Q., Dummer, W., Shen, H., Cebra, J.J., and Surh, C.D. (2005). Recent immune status determines the source of antigens that drive homeostatic T cell expansion. *J Immunol* *174*, 3158-3163.
- Kim, J.M., Rasmussen, J.P., and Rudensky, A.Y. (2007). Regulatory T cells prevent catastrophic autoimmunity throughout the lifespan of mice. *Nat Immunol* *8*, 191-197.
- Leslie, J.L., Vendrov, K.C., Jenior, M.L., and Young, V.B. (2019). The Gut Microbiota Is Associated with Clearance of *Clostridium difficile* Infection Independent of Adaptive Immunity. *mSphere* *4*.
- Min, B., Yamane, H., Hu-Li, J., and Paul, W.E. (2005). Spontaneous and homeostatic proliferation of CD4 T cells are regulated by different mechanisms. *J Immunol* *174*, 6039-6044.
- Reeves, A.E., Koenigsknecht, M.J., Bergin, I.L., and Young, V.B. (2012). Suppression of *Clostridium difficile* in the gastrointestinal tracts of germfree mice inoculated with a murine isolate from the family Lachnospiraceae. *Infect Immun* *80*, 3786-3794.
- Stabler, R.A., Valiente, E., Dawson, L.F., He, M., Parkhill, J., and Wren, B.W. (2010). In-depth genetic analysis of *Clostridium difficile* PCR-ribotype 027 strains reveals high genome fluidity including point mutations and inversions. *Gut Microbes* *1*, 269-276.
- Stockinger, B., Barthlott, T., and Kassiotis, G. (2004). The concept of space and competition in immune regulation. *Immunology* *111*, 241-247.
- Theriot, C.M., Koumpouras, C.C., Carlson, P.E., Bergin, II, Aronoff, D.M., and Young, V.B. (2011). Cefoperazone-treated mice as an experimental platform to assess differential virulence of *Clostridium difficile* strains. *Gut Microbes* *2*, 326-334.
- Ubeda, C., Lipuma, L., Gobourne, A., Viale, A., Leiner, I., Equinda, M., Khanin, R., and Pamer, E.G. (2012). Familial transmission rather than defective innate immunity shapes the distinct intestinal microbiota of TLR-deficient mice. *J Exp Med* *209*, 1445-1456.

REVIEWER COMMENTS

Reviewer #1 (Remarks to the Author):

In this revised manuscript, Littman, et al. present a more detailed view of immune system components that are required for FMT-mediated clearance of *C. difficile*. Using mice that are genetically deficient in B cells or different T cell subsets, they find that CD4+ T cells are important for responsiveness to FMT. They go on to demonstrate that transient depletion of Tregs also abrogates the effectiveness of FMT, concluding that Tregs are critical for engraftment of FMT. However, there are three key points that remain muddled and make the authors' overall conclusion (that Tregs are necessary for engraftment of FMT and normalization of bile acid pools to treat *C. difficile* infection) less convincing.

Specific comments are as follows:

1. It is not clear that the lack of FMT responsiveness in Rag^{-/-} mice is due to the immune defects or to the endogenous microbiota as the data arguing against the latter is not definitive (and is a key element of the manuscript). There is a growing body of literature demonstrating that the endogenous microbiota can limit how well specific species present in donor stool can engraft. Although the authors suggest that there are no differences between the microbiota of Rag(het) and Rag^{-/-} mice at any timepoint, there appears to be a significant difference between the groups at day 38 pi without infection and potentially at d0 as well (Fig 2A, Fig S3B for day 38 without infection). The weighted unifracs also appear to have significant differences at day 38 with infection (Fig S3A). The statistical analysis appears to compare beta-diversity as opposed to a PERMANOVA that assesses whether the overall clustering of the groups are different. The authors attempt to address this issue by transplanting microbiota from *C. difficile*-infected WT mice into antibiotic-treated Rag^{-/-} mice (finding that this still leads to lack of FMT responsiveness). However, they do not demonstrate that this community engrafts "properly" so it is still not clear whether the microbiome at the time of FMT is relevant (and the overarching argument is that the immune system is needed for FMT engraftment which suggests that the Rag^{-/-} mice would not engraft the wt microbiota). In their response to reviewers, the authors provide results from an experiment where they transfer cecal contents from chronically-infected B6 or Rag^{-/-} mice into GF mice and, with a very small sample size of 2-3 mice per group, see no difference in responsiveness to FMT though these mice do not clear the infection as they typically see in their standard experiments (which raises question as to why this difference exists). The authors try to explain why they did not simply transfer wt vs Rag^{-/-} stool into GF mice (or co-house them together), but their rationale for not doing this was not entirely clear. Although these mice would need to be treated with antibiotics to enable *C. difficile* infection, this is the same thing done with the Rag^{-/-} mice in the first place. The question is whether this Rag^{-/-} microbiota does not respond to FMT or whether it is really the immune defect. Treating the ex-GF/Rag microbiota mice with antibiotics more accurately reflects the conditions they are doing throughout the rest of the manuscript.
2. The fact that Treg depletion has to occur well before FMT leaves unexplained the exact role these cells are playing, particularly there is no difference in the number of Tregs between groups at the time of FMT (Fig 4D). Moreover, the finding that depletion of Tregs a few days prior to FMT had no effect (Fig S5F) suggests that Tregs are not directly relevant for FMT engraftment (as suggested by the paper) but is affecting some other process(es) that affects FMT. The identity of this other component remains unclear, but some explanation beyond speculation of why early—but not late—depletion of Tregs has an effect is needed as this is really the crux of the manuscript's message.
3. The data related to bile acids is correlative without actually proving a role for these bile acids (and does not truly "support a role for [secondary] bile acid restoration as one mechanism for resolution of *C. difficile*"). If the idea is that the immune system is necessary for engraftment of the FMT to allow colonization of bile acid converters, will direct provision of these bile acids overcome the defect in immunodeficient mice (Rag and Treg-depleted)?

Minor comments:

1. On page 10, supplementary figure 5G should likely be 5F.
2. For the analysis of microbial species that contribute to FMT success, the authors can likely reduce a lot of the "noise" in their results by also eliminating ASVs that are present in chronically infected wt B6 mice (prior to FMT).
3. Although the authors single out *C. scindens* as one of the 276 ASVs, how many of the other ASVs are also bile acid converters (predicted or known)?

Reviewer #2 (Remarks to the Author):

The authors have responded appropriately and in detail to my comments in their point-by-point response. It is also clear that a lot of additional experiments have been performed that added important data and greatly improved the manuscript.

I have only a couple major conceptual comments left.

The effect in Treg-depleted mice is, although not that surprising, very important for this study. It is very interesting that the Th17 cells don't seem to be the cause for the dysregulation mediated inflammation. It should be discussed whether Th1 cells might be the culprit in the Treg-depleted scenario.

In this context it is even more surprising that Rag1^{-/-} also display increased inflammation as they also lack Treg but of course they also lack T effector cells. This raises the question what innate cell population might be regulated by Treg if the absence of Treg indeed also causes the phenotype in Rag1^{-/-} mice. The authors tried to address this question by performing CD4 or Treg transfer experiments in to Rag1^{-/-} (only shown in response to reviewers). While I am not asking for these technically demanding experiments to be included in the manuscript, a more detailed discussion of this issue would be appropriate.

In addition, since the authors conclude that both Treg as well as bile acid might be involved in the phenotype they observe, I was very surprised by the complete lack of discussion of the recent literature on bile acids and Treg in their Discussion to put their findings into context.

Both recent papers by Hang et al. and Song et al. on the role of bile acids in Treg and Th17 differentiation and homeostasis have to be included in the Discussion to put their findings into the context of the current literature.

References:

Hang, S., Paik, D., Yao, L., Kim, E., Jamma, T., Lu, J., ... Huh, J. R. (2019). Bile acid metabolites control TH17 and Treg cell differentiation. *Nature*, 576(7785), 143–148.

Song, X., Sun, X., Oh, S. F., Wu, M., Zhang, Y., Zheng, W., ... Kasper, D. L. (2020). Microbial bile acid metabolites modulate gut ROR γ ⁺ regulatory T cell homeostasis. *Nature*, 577(7790), 410–415.

Minor comments:

Page 5, line 26: Change 'cross microbial contamination' to 'microbial cross contamination'

Page 9, line 3: Add '.' To Fig. 3d.

Page 12, line 3: Change 'priort' to 'prior'

Reviewer 1:

Major Comments:

Point 1: The Reviewer asks to clarify whether the “lack of FMT responsiveness in Rag^{-/-} mice is due to the immune defects or to the endogenous microbiota”.

The concerns raised by the Reviewer in point 1 are address in detail below. First, we want to clarify the overarching concept of the manuscript to provide a framework for our response. The data in this manuscript provides evidence that the dysregulated immune response elicited in an immunocompromised host (*Rag1^{-/-}*, *C-III^{-/-}*, DT-treated *Foxp3^{DTR}* mice) in the context of *C. difficile* infection prevents a FMT from engrafting. Text in our previous submission used the phrase “FMT-treated *Rag1^{-/-}* mice” to described *C. difficile* infected *Rag1^{-/-}* mice that received a FMT. We believe this terminology was confusing and insufficiently described the mouse cohorts being studied. We have edited the manuscript to denote mice as “*C. difficile* infected FMT-treated mice” to clarify this distinction. Further, new experimental data included in this resubmission demonstrates that uninfected *Rag1^{-/-}* mice are capable of engrafting an FMT, highlighting the difference in FMT receptiveness between infected and uninfected immunodeficient hosts (**Suppl. Fig. 11**). The overall capacity of immunocompromised hosts outside the context of *C. difficile* infection to engraft a FMT is an important question but is beyond the scope of this manuscript.

Point 1A: Reviewer 1 ask to reanalyze our 16s rRNA dataset comparing the microbiota of *Rag1^{HET}* vs *Rag1^{-/-}* mice using new analytical techniques. Specifically the Reviewer states “...there appears to be a significant difference between the groups at day 38 pi without infection and potentially at d0 as well (Fig 2A, Fig S3B for day 38 without infection). The weighted unifrac also appears to have significant differences at day 38 with infection (Fig S3A). The statistical analysis appears to compare beta-diversity as opposed to a PERMANOVA that assesses whether the overall clustering of the groups are different.”

We have reanalyzed our original 16S rRNA dataset and ran PERMANOVA comparing the microbiota of *Rag1^{HET}* vs *Rag1^{-/-}* mice prior to antibiotics, at the day of infection and at the timepoint of FMT (day 36 p.i.). Further, we have included analysis from a second validation dataset of 16S rRNA sequences from naive, antibiotic-treated uninfected and *C. difficile*-infected *Rag1^{HET}* and *Rag1^{-/-}* mice and ran identical statistical analyses.

The 16S rRNA analyses presented in this manuscript test the null hypothesis that there is no difference between groups. These new analyses are now described on Pg. 7, Ln. 3-23, and in **Fig. 2, Suppl. Figs. 3,4, Suppl. Table 1,2** and are summarized below:

1. PERMANOVA statistical tests using weighted or unweighted UniFrac distance did not identify a statistically significant difference between the microbial communities of *C. difficile*-infected *Rag1^{HET}* and *Rag1^{-/-}* mice at the time of FMT in either the original dataset or validation dataset (**Suppl. Table 1,2**)
2. PERMANOVA statistical tests using weighted or unweighted UniFrac distance did not identify a statistically significant difference between the microbial communities of antibiotic-treated *Rag1^{HET}* and *Rag1^{-/-}* mice at day 0 of infection or day 32 post mock infection (**Suppl. Table 1,2**).
3. A PERMANOVA statistical test using unweighted UniFrac distance did identify a statistically significant difference between the microbial communities of pre-antibiotic treated *Rag1^{HET}* and *Rag1^{-/-}* mice (**Suppl. Table 1**).

We do not wish to over-interpret our microbiota data and have conferred with Dr. Kyle Bittinger, co-author on this manuscript and Analytics Core Director of the Microbiome Center at the Children’s Hospital of Philadelphia on how to describe this data in the Results. Based on his previous experience with similar datasets (Bittinger et al., 2020; Bolyen et al., 2019; Feres et al., 2020; Sinha et al., 2020), we have adjusted our description of these data to state that our microbial community data analyses do not lead us to reject the

null hypothesis (i.e. – our analyses do not provide evidence that there is a difference between the microbiota of *C. difficile*-infected *Rag1*^{HET} and *Rag1*^{-/-} mice).

The analytical approaches used in this manuscript are robust enough to identify microbiota differences. For example, we are able to reject the null hypothesis when comparing the microbiota of *C. difficile*-infected FMT-treated *Rag1*^{HET} mice to the microbiota of *C. difficile*-infected FMT-treated *Rag1*^{-/-} mice using multiple methods of analysis (**Fig. 5, Suppl. Fig. 8 Suppl. Table 3**). These same analyses, however, did not lead us to reject the null hypothesis comparing the microbiota of *C. difficile* infected *Rag1*^{HET} mice to the microbiota of *C. difficile* infected *Rag1*^{-/-} mice in either our primary or validation cohort.

Korn LL et. al. reported that naive *Rag1*^{HET} and *Rag1*^{-/-} mice do have subtle but significant differences in intestinal microbial communities (Korn et al., 2014). Our data also identifies a statistically significant difference between the microbiota of naive *Rag1*^{HET} and *Rag1*^{-/-} mice (**Suppl. Table 1**), however this difference is no longer detected at the time of FMT.

In new experimental data, we observed no defect in FMT engraftment in antibiotic-treated *Rag1*^{-/-} mice that are not infected with *C. difficile* (**Suppl. Fig. 11**) Therefore the focus of the manuscript is on the microbial community composition at the time of FMT in *C. difficile* infected hosts, as this is the environment in which the bacteria from the FMT inoculum is attempting to engraft and we observe a differential success rate in engraftment between mouse genotypes.

Point 1B: The Reviewer states, “The authors attempt to address this issue by transplanting microbiota from *C. difficile*-infected WT mice into antibiotic-treated *Rag1*^{-/-} mice (finding that this still leads to lack of FMT responsiveness). However, they do not demonstrate that this community engrafts “properly” so it is still not clear whether the microbiome at the time of FMT is relevant.”

We thank the Reviewer for the suggestion and we have now included the transplanting microbiota from *C. difficile* infected C57BL/6 and *Rag1*^{-/-} mice in the PCoA plots in **Supplemental Figure 5** and **Reviewer Fig 1C** to visualize community engraftment. The experiments that transfer the cecal content of *C. difficile*-infected C57BL/6 or *Rag1*^{-/-} mice into antibiotic-treated *Rag1*^{-/-}, *Rag1*^{HET}, or germ-free C57BL/6 recipient mice (**Suppl. Fig. 5, Reviewer Fig. 1**) address whether the microbiota of a *C. difficile* infected *Rag1*^{-/-} mice at the time of FMT independently drives FMT failure regardless of the host's immune status. C57BL/6 mice that receive cecal content from *C. difficile*-infected *Rag1*^{-/-} mice still do respond to a FMT. Therefore, the FMT failure phenotype of *C. difficile* infected *Rag1*^{-/-} mice cannot be adoptively transferred by microbiota transplantation.

Reviewer Figure 1. GF B6 mice transplanted with microbiome derived from C57BL/6 or *Rag1*^{-/-} mice exhibit equivalent resolution of *C. difficile* following FMT. (A) Experimental schematic. Antibiotic-treated C57BL/6 and *Rag1*^{-/-} were infected with *C. difficile*. At day 30 p.i. cecal content (containing *C. difficile*) was transferred into reciprocal gnotobiotic (GF) C57BL/6 mice. At day 21 post cecal transplant, mice were administered FMT and (B) *C. difficile* burden monitored in the fecal pellets. Ex-GF B6 (B6 Microbiome) (C) UniFrac principal coordinate analysis of 16S bacterial rRNA sequence reads from the fecal pellets of mice post FMT. Each PCoA plot represents a timecourse of an individual mouse. Green squares represent FMT source. Dark Red or blue circles represent the original microbiota donor. n =3. Ex-GF B6 (*Rag1*^{-/-} Microbiome) n =2.

Point 1C: The Reviewer states, “...the overarching argument is that the immune system is needed for FMT engraftment which suggests that the Rag1^{-/-} mice would not engraft the wt microbiota.”

We would like to take this opportunity to clarify a potential point of confusion that may have caused a misinterpretation on the overarching argument of the manuscript. Our data supports that Rag1^{-/-} mice fail to engraft a FMT in the context of both a *C. difficile* infection and the subsequent immune response elicited by the infection. Based on our data, we hypothesize that uninfected Rag1^{-/-} mice would be capable of FMT engraftment if there was no *C. difficile* infection induced immune activation. We experimentally tested this hypothesis by administering a FMT to mock infected Rag1^{-/-} mice that underwent the same antibiotic treatment regimen as *C. difficile* infected mice. 16S rRNA microbial profiling of the feces from antibiotic treated, uninfected Rag1^{-/-} mice prior to and following FMT demonstrate successful engraftment. Post FMT the microbiota of antibiotic treated, uninfected Rag1^{-/-} mice clustered most similarly to the FMT inoculum as determined by both weighted and unweighted UniFrac analysis. These new analyses are now described on Pg. 12, Ln. 18-24, and in **Suppl. Fig. 11**.

Point 1D: The Reviewer notes data present in Reviewer figure 1 has a small sample size and asks why the ex-germfree mice in this experiment do not completely resolve *C. difficile* infection as observed in other experiments.

We agree the incomplete resolution of infection is noteworthy, though it is not entirely unexpected and these results do not contradict our overarching model. First, we observed a higher than normal mortality rate in the acute phase of infection in ex-germfree (ex-GF) mice and therefore had a small sample size to conduct the subsequent FMT. Second, we hypothesize the ex-GF C57BL/6 mice that survived the acute phase of infection experienced increased and prolonged inflammation in response to *C. difficile* infection compared to specific pathogen free (SPF) C57BL/6 mice that hindered subsequent complete *C. difficile* resolution following FMT. GF mice have several systemic and intestinal immune deficiencies that leave them more susceptible to infection (Smith et al., 2007). Therefore the immune response and subsequent persisting inflammation in ex-GF mice infected with *C. difficile* would not be predicted to fully replicate a SPF wild type mouse. For example, GF mice have a diminished colonic T_{reg} cell compartment that lack commensal induced T_{reg} cells (Atarashi et al., 2011). Therefore, it is plausible that the partially defective T_{reg} cell compartment may contribute incomplete resolution in the ex-GF mice, however more experimental data beyond the scope of this manuscript is needed to definitively draw this conclusion.

Point 1E: The Reviewer states, “The authors try to explain why they did not simply transfer wt vs Rag1^{-/-} stool into GF mice (or co-house them together), but their rationale for not doing this was not entirely clear.

The rationale of transferring cecal content from *C. difficile* infected C57BL/6 or Rag1^{-/-} mice into ABX-treated or GF mice is to test whether the microbial communities present at the day of FMT impact FMT engraftment and *C. difficile* resolution independent of the immune activation status. We agree with the Reviewer that “There is a growing body of literature demonstrating that the endogenous microbiota can limit how well specific species present in donor stool can engraft.” The proposed experiment of cohousing GF mice with Rag1^{-/-} or Rag1^{HET} mice, treating them with antibiotics, infecting the ex-GF mice with *C. difficile* and then administering an FMT to test the contribution of the original microbiota to FMT success is an interesting and distinct question. In new experimental data, we have conducted this experiment and observed that *C. difficile* infected ex-GF mice cohoused with Rag1^{-/-} or Rag1^{HET} mice prior to infection up to the day of FMT are equally capable of resolving *C. difficile* infection (**Reviewer Fig. 2**). This data provides evidence that the microbiota from naive Rag1^{-/-} mice does not inherently imprint FMT failure in immunocompetent mice. We believe this data may sidetrack readers from the core theme of the manuscript on role of the immune system at the time of FMT in supporting resolution of *C. difficile* infection, therefore, we have not include this data in the manuscript resubmission. However, we will include this data at the Reviewer’s request.

Reviewer Figure 2. Germfree B6 mice colonized with the microbiota of naïve $Rag1^{-/-}$ mice exhibit no defect in FMT mediated resolution of *C. difficile* infection. **(A)** Experimental schematic. Germfree (GF) C57BL/6 mice were cohoused with $Rag1^{HET}$ or $Rag1^{-/-}$ mice for 21 days prior to the start of antibiotic treatment. Following ABX treatment, ex-GF B6, $Rag1^{HET}$ or $Rag1^{-/-}$ mice were infected with *C. difficile*. At day 21 p.i. ex-GF B6 mice were administered an FMT and singly housed. **(B)** *C. difficile* burden in the fecal pellets following FMT. Ex-GF B6 cohoused with $Rag1^{HET}$ mice n=6. Ex-GF B6 cohoused with $Rag1^{-/-}$ mice n=5.

Point 1F: The Reviewer asks “whether this $Rag^{-/-}$ microbiota does not respond to FMT or whether it is really the immune defect”

We observed *C. difficile* infected $Rag1^{-/-}$ mice fail to engraft a FMT and subsequently do not resolve *C. difficile* infection. In a series of supporting experiments we address the question of whether the microbiota of $Rag1^{-/-}$ mice inherently inhibits FMT engraftment independent of immune defects. We conclude that the microbiota of $Rag1^{-/-}$ mice does not independently reject FMT engraftment. FMT failure observed in $Rag1^{-/-}$ mice is driven by *C. difficile* infection induced immune activation. We make this conclusion based on the following observations:

1. FMT is successful in ABX-treated, uninfected $Rag1^{-/-}$ mice (**Suppl. Fig. 11**).
 - a. Conclusion A: $Rag1^{-/-}$ mice microbiota is not inherently refractory to FMT.
2. FMT is successful in ABX-treated C57BL/6 mice receiving cecal content from *C. difficile* infected $Rag1^{-/-}$ mice. (**Suppl. Fig. 5**).
 - a. Conclusion B: The microbiota from *C. difficile* infected $Rag1^{-/-}$ mice adoptively transferred into a wild-type mouse does not inherently inhibit FMT success.
3. FMT is unsuccessful in ABX-treated $Rag1^{-/-}$ mice receiving cecal content from *C. difficile* infected $Rag1^{HET}$ mice. (**Suppl. Fig. 5**).
 - a. Conclusion C: The microbiota from *C. difficile* infected wild-type mice adoptively transferred into a $Rag1^{-/-}$ mouse does not inherently support FMT success.

Point 2A: The Reviewer asks “what is the role T_{reg} cells have in FMT-mediated resolution of *C. difficile* infection since T_{reg} cell depletion occurs well before FMT and there is no difference in the number of T_{reg} cells between DT treated and PBS treated *C. difficile* infected mice.”

Our data supports the model where T_{reg} cells indirectly support FMT engraftment through regulation of intestinal inflammation. T_{reg} cells control intestinal inflammation and it is the inflammatory environment that is refractory to FMT engraftment.

While there is no difference in the total number of T_{reg} cells at time of FMT, the proportion of T_{reg} cells within the $CD4^+$ T cell compartment is significantly reduced. This decreased frequency is due to expansion of other $CD4^+$ T helper cell subsets, specifically, T_H17 and T_H1 cells (**Fig. 4H**). Further, DT-treated *C. difficile* infected Foxp3-DTR mice have increased infiltration of $Ly6c^+$ inflammatory monocytes and $Ly6g^+$ neutrophils into the lamina propria at the day of FMT (day 21 p.i.) (**Fig. 4I**). These new data, in combination with data showing increased expression of proinflammatory immune response genes (**Fig. 4J**) demonstrate that the returning T_{reg} cell population is insufficient at regulating the intestinal inflammatory environment of *C. difficile*

infected mice. We have now added **Fig. 4F,H,I** to better visualize the increased inflammation at the time of FMT and have described these results on Pg. 10-11, Ln 23-26; 1-9.

Point 2B: The Reviewer states, “...the finding that depletion of Tregs a few days prior to FMT had no effect (Fig S5F) suggests that Tregs are not directly relevant for FMT engraftment (as suggested by the paper) but is affecting some other process(es) that affects FMT.”

T_{reg} cell depletion 1-2 days prior to FMT may not be sufficient time to drive the inflammatory conditions that inhibit an FMT. Indeed Kim JM et al found that transient depletion of T_{reg} cells led to colitis in 7-10 days following initial DT treatment despite restoration of the T_{reg} cell population by this timepoint (Kim et al., 2007). In new experimental data, we demonstrate the returning T_{reg} cell population is not required to directly inhibit FMT engraftment. Instead, the lack of T_{reg} cell-mediated immunoregulation indirectly results in FMT failure. Sustained DT treatment to maintain depletion of intestinal T_{reg} cells starting at day 8 p.i. and continuing through day 21 p.i. (day 0 FMT) also impaired FMT-mediated resolution of *C. difficile* infection. This data is described on Pg. 11, Ln. 10-15 and in **Suppl. Fig. 7E**.

Point 3A: The Reviewer states, “The data related to bile acids is correlative without actually proving a role for these bile acids...”

We agree with the Reviewer that the data presented in this manuscript does not demonstrate that secondary bile acid restoration drives resolution of *C. difficile* infection. We do not wish to make this claim and agree with the Reviewer the need to clarify our metabolite data. First, we have changed the nomenclature of the manuscript from using “microbiome”, which can be defined as all microbial organisms and microbial-derived metabolites that reside within a mammalian host, to “microbiota”, which is more specific to the community of microbial organisms. We believe this more conservative nomenclature is more appropriate to describe the findings of this manuscript.

In addition, to complement to the 1^o and 2^o bile acid data presented in the original submission, we have included additional new metabolite data by conducting a targeted metabolite screen for short chain fatty acids and amino acids in the cecal content of *Rag1*^{HET} and *Rag1*^{-/-} mice at steady-state, during *C. difficile* infection and following FMT. No statistical significant difference was observed in the SCFA assessed while some amino acids, such as valine and leucine were modestly increased in FMT-treated *Rag1*^{HET} mice compared to FMT-treated *Rag1*^{-/-} mice. These new data are now presented in **Fig. 6A-C, Suppl. Fig. 12A,B** and discussed on Pg. 13, Ln. 6-17.

The most profound differences observed in our targeted metabolite screen were 2^o bile acid levels between FMT-treated *C. difficile* infected *Rag1*^{HET} mice and FMT-treated *C. difficile* infected *Rag1*^{-/-} mice (**Fig. 6A,C, Suppl. Fig. 12D**). Previous clinical reports have shown that 2^o bile acids are reduced in *C. difficile* infected patients and are restored following successful FMT (Seekatz et al., 2018; Weingarden et al., 2016). As such, 2^o bile acids serve as important metabolite biomarkers that can indicate whether an FMT has successfully engrafted. Therefore we use 2^o bile acids as representative examples to support evidence that unsuccessful FMT engraftment results in downstream differences in intestinal metabolite composition. We believe presenting this data is beneficial to clinicians and scientists that are familiar with the 2^o bile acid literature in the context of *C. difficile* infection and therefore have elaborated on this class of metabolites in the Discussion. However, 2^o bile acids are not the only metabolites that are not restored in FMT non-responsive hosts and have adjusted the text of our manuscript to clarify our observations involving of secondary bile acid restoration on Pg. 14, Ln 2-3; Pg. 15-16, Ln. 21-25, 1-2.

Point 3B: The Reviewer asks if direct administration of 2^o bile acids in *C. difficile* infected *Rag1*^{-/-} mice will promote resolution of *C. difficile* infection.

There are likely several metabolites that are needed in combination promote an intestinal environment inhospitable to *C. difficile*. Therefore we hypothesize direct administration of 2^o bile acids does not resolve *C. difficile* in either a fully immunocompetent or immunodeficient host. To test this hypothesis, *C. difficile* infected *Rag1*^{HET} and *Rag1*^{-/-} mice were treated with deoxycholic acid (DCA) and lithocholic acid (LCA) via

oral gavage and in the drinking water for 10 days (as described in (Song et al., 2020)) and *C. difficile* burden assessed. DCA and LCA were assessed because both metabolites have been found to inhibit *C. difficile* growth *in vitro* and were the two most differently abundant 2° bile acids in FMT-treated *Rag1^{HET}* mice compared to FMT-treated *Rag1^{-/-}* mice. DCA and LCA administration did not reduced *C. difficile* burden in either *Rag1^{HET}* and *Rag1^{-/-}* mice demonstrating that these two bile acids were not sufficient to resolve *C. difficile* infection in our mouse model (**Reviewer Fig. 3**).

Secondary bile acids come in several forms of conjugated and iso-bile acid derivatives. The metabolite screen presented in our manuscript was limited and assessed the concentration of 15 bile acid. Thus we can not rule out the possibility of some combinations of 2° bile acids derivatives can directly resolve *C. difficile* infection *in vivo*. Determining if such a combination exists will require extensive *in vitro* and *in vivo* studies that are beyond the scope of this manuscript.

Reviewer Figure 3. Deoxycholic acid and lithocholic acid administration does not resolve *C. difficile* infection in either *Rag1^{HET}* or *Rag1^{-/-}* mice **A)** Experimental schematic. *Rag1^{HET}* or *Rag1^{-/-}* mice were treated with antibiotics and infected with *C. difficile*. At day 21 p.i. mice were switched to drinking water containing 0.004% LCA and 0.01% DCA. Mice were also gavaged every other day with 250 ul of 0.04% LCA and 0.1% DCA. **(B)** *C. difficile* burden in the fecal pellets following start of DCA/LCA treatment. *Rag1^{HET}* mice n=7. *Rag1^{-/-}* mice n=5.

Minor Comments

1. The Reviewer states, “On page 10, supplementary figure 5G should likely be 5F.”

This has been corrected, thank you.

2. The Reviewer states, “For the analysis of microbial species that contribute to FMT success, the authors can likely reduce a lot of the “noise” in their results by also eliminating ASVs that are present in chronically infected wt B6 mice (prior to FMT).”

Thank you, the heatmap presented in **Supplemental Figure 10** does use these criteria. The criteria for ASVs that contribute to FMT are the following:

- (1) present in the FMT inoculum,
- (2) absent in *C. difficile* infected *Rag1^{HET}* or *Rag1^{-/-}* mice prior to FMT,
- (3) absent in PBS-treated *Rag1^{HET}* or *Rag1^{-/-}* mice,
- (4) absent in FMT-treated *Rag1^{-/-}* mice,
- (5) present in FMT-treated *Rag1^{HET}* mice.

We have modified the text of the manuscript on Pg. 12, Ln 11-15 to clarify to point.

3. The Reviewer asks “how many of the other ASVs [besides *C. scindens*] are also bile acid converters (predicted or known)?”

We thank the reviewer for bring up this point. We analyzed our dataset for ASV that shared > 96% homology to *C. hylemonae* and *C. hiranonis* two other known bile acid converters (Reed et al., 2020). *C.*

hiranonis was not identified in any samples. *C. hylemonae* was identified in small abundance in the FMT inoculum however, it was not observed in FMT treated *Rag1*^{HET} or *Rag1*^{-/-} mice. This data is presented in **Reviewer Fig. 4**.

Reviewer Figure 4. Secondary bile acid producer *C. hylemonae* in FMT inoculum and *Rag1*^{HET} or *Rag1*^{-/-} mice before and after FMT. ASVs that share > 96% sequence homology to *C. hylemonae* 16S rRNA sequence reads.

Reviewer 2

General Comments:

Reviewer 2 stated that our resubmission responded appropriately and in detail to the reviewer's comments and the new experimental data added in the resubmission was "important" and "greatly improved the manuscript". This reviewer had a few conceptual comments to be addressed.

Major Comments:

1. The Reviewer asks to discuss the role of T_H1 cells following T_{reg} cell depletion in *C. difficile* infected mice.

We agree with the Reviewer that the T_H1 population is of equal interest as T_{reg} cells and T_H17 cells and we have added two pieces of new experimental data to address the role of T_H1 cells in FMT-mediated clearance of *C. difficile* infection. (1) T-bet deficient mice (*Tbx21*^{-/-}) were infected with *C. difficile* and treated with FMT to assess the capacity to resolve *C. difficile* infection. *Tbx21*^{-/-} mice resolved *C. difficile* infection following FMT indicating that CD4⁺ T_H1 cells are not necessary to support FMT mediated resolution of *C. difficile* infection. This data is now described on Pg. 9-10, Ln. 26; 1-2 and display in **Suppl. Fig. 7C**. (2) We also analyzed CD4⁺ T cell subsets in Foxp3-DTR mice following DT treatment and observe increased frequency and total number of T_H1 and T_H17 cells in the lamina propria of the large intestine of DT treated mice at the time of FMT (**Fig 4H,I**). This data in combination with increased expression of type 1 and type-17 cytokines (**Fig. 4J**) support a model where T_H1 and T_H17 cells cellular subsets contribute to the intestinal inflammatory environment following loss of the T_{reg} cell compartment. This data is discussed on Pg. 10-11, Ln 22-25, 1-3 and data displayed in **Fig. 4H, I**.

2. The Reviewer asks, "...what innate cell population might be regulated by Treg.?"

In addition to expansion of T_H1 and T_H17 cells in DT treated *C. difficile* infected Foxp3-DTR mice we also observed increased infiltration of inflammatory innate immune cells, specifically Ly6c⁺ inflammatory monocytes and Ly6g⁺ neutrophils, into the large intestine lamina propria compared to PBS-treated *C. difficile* infected Foxp3-DTR mice and DT treated uninfected Foxp3-DTR mice. This data is now described on Pg. 11, Ln 1-3 in the manuscript and displayed in **Fig. 4I**. *Rag1*^{-/-} and C-Il^{-/-} mice induce an acute innate inflammatory response that is critical for survival following *C. difficile* infection (Abt et al., 2015; Johnston et al., 2014). However, this inflammatory response is not subsequently regulated by an intact T_{reg} cell population. The resulting chronic intestinal inflammation (**Fig. 2E,F, 4J**) is sufficient to prevent acute mortality following *C. difficile* infection, but promotes an intestinal environment that is refractory to FMT engraftment.

3. The Reviewer asks to discuss recent findings on the role of bile acids in Treg and Th17 differentiation and homeostasis in the context of the manuscript

We thank the reviewer for identifying this gap in our Discussion.

Three recent papers report that 2° bile acid derivatives can promote T_{reg} cell development in the colon via direct signaling on T_{reg} cells through the Vitamin D Receptor or the CNS3 enhancer or indirectly via FXR signaling on DCs (Campbell et al., 2020; Hang et al., 2019; Song et al., 2020). Our data demonstrate low to absent 2° bile acid pools in *C. difficile* infected mice prior to FMT indicating that 2° bile acids are likely not inducing peripheral T_{reg} cell development at the time of FMT. Following FMT, restoration of 2° bile acids could drive a potential positive feed forward loop that promotes T_{reg} cell expansion, further reducing intestinal inflammation and enabling repopulation of inflammation sensitive commensal bacteria. The timing and context of this potential feed forward mechanism will require further study.

We have now included this discussion on bile acids and T_{reg} cell development on Pg. 16 Ln. 3-16 and we believe this substantially improves the Discussion section.

Minor Comments

1. The Reviewer states, “Page 5, line 26: Change ‘cross microbial contamination’ to ‘microbial cross contamination’”.

Thank you, corrected.

2. The Reviewer states, “Page 9, line 3: Add ‘.’ To Fig. 3d.”.

Thank you, corrected.

3. The Reviewer states, “Page 12, line 3: Change ‘priot’ to ‘prior’”. –

Thank you, corrected.

References:

- Abt, M.C., Lewis, B.B., Caballero, S., Xiong, H., Carter, R.A., Susac, B., Ling, L., Leiner, I., and Pamer, E.G. (2015). Innate Immune Defenses Mediated by Two ILC Subsets Are Critical for Protection against Acute *Clostridium difficile* Infection. *Cell Host Microbe* 18, 27-37.
- Atarashi, K., Tanoue, T., Shima, T., Imaoka, A., Kuwahara, T., Momose, Y., Cheng, G., Yamasaki, S., Saito, T., Ohba, Y., *et al.* (2011). Induction of colonic regulatory T cells by indigenous *Clostridium* species. *Science* 331, 337-341.
- Bittinger, K., Zhao, C., Li, Y., Ford, E., Friedman, E.S., Ni, J., Kulkarni, C.V., Cai, J., Tian, Y., Liu, Q., *et al.* (2020). Bacterial colonization reprograms the neonatal gut metabolome. *Nat Microbiol* 5, 838-847.
- Bolyen, E., Rideout, J.R., Dillon, M.R., Bokulich, N.A., Abnet, C.C., Al-Ghalith, G.A., Alexander, H., Alm, E.J., Arumugam, M., Asnicar, F., *et al.* (2019). Reproducible, interactive, scalable and extensible microbiome data science using QIIME 2. *Nat Biotechnol* 37, 852-857.
- Campbell, C., McKenney, P.T., Konstantinovskiy, D., Isaeva, O.I., Schizas, M., Verter, J., Mai, C., Jin, W.B., Guo, C.J., Violante, S., *et al.* (2020). Bacterial metabolism of bile acids promotes generation of peripheral regulatory T cells. *Nature* 581, 475-479.
- Feres, M., Retamal-Valdes, B., Fermiano, D., Faveri, M., Figueiredo, L.C., Mayer, M., Lee, J.J., Bittinger, K., and Teles, F. (2020). "Microbiome changes in young periodontitis patients treated with adjunctive metronidazole and amoxicillin". *J Periodontol*.
- Hang, S., Paik, D., Yao, L., Kim, E., Trinath, J., Lu, J., Ha, S., Nelson, B.N., Kelly, S.P., Wu, L., *et al.* (2019). Bile acid metabolites control TH17 and Treg cell differentiation. *Nature* 576, 143-148.

Johnston, P.F., Gerding, D.N., and Knight, K.L. (2014). Protection from *Clostridium difficile* infection in CD4 T Cell- and polymeric immunoglobulin receptor-deficient mice. *Infect Immun* 82, 522-531.

Kim, J.M., Rasmussen, J.P., and Rudensky, A.Y. (2007). Regulatory T cells prevent catastrophic autoimmunity throughout the lifespan of mice. *Nat Immunol* 8, 191-197.

Korn, L.L., Thomas, H.L., Hubbeling, H.G., Spencer, S.P., Sinha, R., Simkins, H.M., Salzman, N.H., Bushman, F.D., and Laufer, T.M. (2014). Conventional CD4+ T cells regulate IL-22-producing intestinal innate lymphoid cells. *Mucosal Immunol* 7, 1045-1057.

Reed, A.D., Nethery, M.A., Stewart, A., Barrangou, R., and Theriot, C.M. (2020). Strain-Dependent Inhibition of *Clostridioides difficile* by Commensal *Clostridia* Carrying the Bile Acid-Inducible (bai) Operon. *J Bacteriol* 202.

Seekatz, A.M., Theriot, C.M., Rao, K., Chang, Y.M., Freeman, A.E., Kao, J.Y., and Young, V.B. (2018). Restoration of short chain fatty acid and bile acid metabolism following fecal microbiota transplantation in patients with recurrent *Clostridium difficile* infection. *Anaerobe* 53, 64-73.

Sinha, S.R., Haileselassie, Y., Nguyen, L.P., Tropini, C., Wang, M., Becker, L.S., Sim, D., Jarr, K., Spear, E.T., Singh, G., *et al.* (2020). Dysbiosis-Induced Secondary Bile Acid Deficiency Promotes Intestinal Inflammation. *Cell Host Microbe* 27, 659-670 e655.

Smith, K., McCoy, K.D., and Macpherson, A.J. (2007). Use of axenic animals in studying the adaptation of mammals to their commensal intestinal microbiota. *Semin Immunol* 19, 59-69.

Song, X., Sun, X., Oh, S.F., Wu, M., Zhang, Y., Zheng, W., Geva-Zatorsky, N., Jupp, R., Mathis, D., Benoist, C., and Kasper, D.L. (2020). Microbial bile acid metabolites modulate gut RORgamma(+) regulatory T cell homeostasis. *Nature* 577, 410-415.

Weingarden, A.R., Dosa, P.I., DeWinter, E., Steer, C.J., Shaughnessy, M.K., Johnson, J.R., Khoruts, A., and Sadowsky, M.J. (2016). Changes in Colonic Bile Acid Composition following Fecal Microbiota Transplantation Are Sufficient to Control *Clostridium difficile* Germination and Growth. *PLoS One* 11, e0147210.

REVIEWERS' COMMENTS

Reviewer #1 (Remarks to the Author):

In this revised manuscript, Littman, et al., have responded adequately to all of my concerns and I have no substantive comments remaining. There are 2 minor points below that I think should get amended prior to publication.

1. I strongly encourage the authors to include the data included as Reviewer Figure 2 into the main manuscript. This is the most compelling data offered that it is the immune defect—and not just differences in the microbiota—that drive engraftment/effects of FMT.
2. Lines 9-10 on page 12 (“It is likely that a consortium of microbial species act in concert to reshape the intestinal environment”) should get removed. The authors offer no evidence to support whether a consortium of microbes or a single microbe is necessary. Many other studies have similarly found a large number of differentially abundant taxa between phenotypes of interest but were ultimately able to localize the effects to a single taxon.

Reviewer #2 (Remarks to the Author):

The authors have adequately addressed my comments and even added more additional data.

There are quite a few formatting issues with newly added references and I don't think the pages and lines referred to in the response to reviewers are correct:

- 1) some newly added references have not been correctly formatted, see eg:
 - page 15 {Song, 2020 #1044}{Campbell, 2020 #1042}{Hang,2019 #950}
 - page 21 {Friedman, 2018 #1066;Ramsteijn, 2020 #1067}
 - age 22 {Friedman, 2018 #1066;Ramsteijn, 2020 #1067}

This likely results in a from Bibliography list.

- 2) It was hard for me to match up the page and one references given in the response to reviewers to the actual mansucrtipot. I don't think these refer to correct pages and lines in the manuscript.

Revised manuscript submission:

Littman et al. Host immunity modulates efficacy of microbiome transplantation in treatment of *Clostridioides difficile* infection.

Reviewer 1:

1. I strongly encourage the authors to include the data included as Reviewer Figure 2 into the main manuscript. This is the most compelling data offered that it is the immune defect—and not just differences in the microbiota—that drive engraftment/effects of FMT.

We have included Reviewer Figure 2 the revised manuscript. This data is now part of Supplemental Figure 5E,F and is described in the body of the manuscript on Pg. 8 Ln. 4-11.

2. Lines 9-10 on page 12 (“It is likely that a consortium of microbial species act in concert to reshape the intestinal environment”) should get removed. The authors offer no evidence to support whether a consortium of microbes or a single microbe is necessary. Many other studies have similarly found a large number of differentially abundant taxa between phenotypes of interest but were ultimately able to localize the effects to a single taxon.

This sentence has been removed from the manuscript.

Reviewer 2

1. Some newly added references have not been correctly formatted, see eg:

- page 15 {Song, 2020 #1044}{Campbell, 2020 #1042}{Hang,2019 #950}

- page 21 {Friedman, 2018 #1066;Ramsteijn, 2020 #1067}

- age 22 {Friedman, 2018 #1066;Ramsteijn, 2020 #1067}

This likely results in a from Bibliography list.

These errors have been corrected in the revised manuscript.